# Pervasive translation of circular RNAs driven by short IRES-like elements

Xiaojuan Fan [1,2,4], Yun Yang[1,3,4], Chuyun Chen [1,4] & Zefeng Wang [1,2✉]

Some circular RNAs (circRNAs) were found to be translated through IRES-driven mechanism, however the scope and functions of circRNA translation are unclear because endogenous IRESs are rare. To determine the prevalence and mechanism of circRNA translation, we develop a cell-based system to screen random sequences and identify 97 overrepresented hexamers that drive cap-independent circRNA translation. These IRES-like short elements are significantly enriched in endogenous circRNAs and sufficient to drive circRNA translation. We further identify multiple *trans*-acting factors that bind these IRES-like elements to initiate translation. Using mass-spectrometry data, hundreds of circRNA-coded peptides are identified, most of which have low abundance due to rapid degradation. As judged by mass-spectrometry, 50% of translatable endogenous circRNAs undergo rolling circle translation, several of which are experimentally validated. Consistently, mutations of the IRES-like element in one circRNA reduce its translation. Collectively, our findings suggest a pervasive translation of circRNAs, providing profound implications in translation control.

[1] Bio-med Big Data Center, CAS Key Laboratory of Computational Biology, CAS Center for Excellence in Molecular Cell Science, Shanghai Institute of Nutrition and Health, Shanghai, China. [2] University of Chinese Academy of Sciences, Chinese Academy of Sciences, Beijing, China. [3] Present address: CirCode BioMedicine, Pudong, Shanghai, China. [4] These authors contributed equally: Xiaojuan Fan, Yun Yang, Chuyun Chen. ✉email: wangzefeng@picb.an.cn

Circular RNAs (circRNAs) have recently been demonstrated as a class of abundant and conserved RNAs in animals and plants (for review, see[1–3]). Most circRNAs are produced from back-splicing of pre-mRNAs, and are predominantly localized in cytoplasm[3–5]. However, the general function of circRNA in vivo is still an open question. Several circRNAs have been found as molecular sponges to sequester miRNAs[6,7] or RNA binding proteins (RBPs)[8] (i.e., as competitors of the linear mRNAs), whereas some nuclear circRNAs were reported to promote transcription of nearby genes[9,10]. Since in vitro synthesized circRNAs can be translated in cap-independent fashion[11] and most circRNAs are localized in cytoplasm, it is highly possible that many circRNAs function as mRNAs to direct protein synthesis.

Recently we and other groups reported that circRNAs can indeed be translated in vivo via different internal ribosome entry sites (IRESs)[12–15]. Because circRNAs lack a 5′ end, its translation can only be initiated in a cap-independent mechanism that requires the internal ribosomal entry site (IRES). However the endogenous IRESs are infrequent in eukaryotic transcriptomes, and even their existence is sometimes under debate[16–18], casting doubts on the scope of circRNA translation. In support of this notion, a recent study has identified hundreds of putative IRESs by systematically searching selected viral sequences and 5′-UTR of human mRNA[19]; however, only a small fraction (<1.5%) of ~100,000 known circRNAs[20] contain these newly identified IRESs.

To study the scope of circRNA translation, we developed a cell-based reporter system to screen a random library of short sequences that drive circRNA translation. Through a near-saturated screen and subsequent bioinformatics analyses, we identified 97 IRES-like hexamers that are clustered into 11 groups with AU-rich consensus motifs. The IRES-like elements are significantly enriched in human circRNAs compared to all linear RNAs, suggesting that they are positively selected in circRNAs. Since these IRES-like hexamers account for ~2% of all hexamers (97/4096), any sequences longer than 50-nt should contain such an element by chance, implying that most circRNAs in human cells can potentially be translated through the IRES-like short elements. Consistently, circRNAs containing only the coding sequences can indeed be translated in a rolling circle fashion, presumably from internal IRES-like short elements in the coding region. We further identified hundreds of circRNA-coded proteins with mass spectrometry datasets, and explored the potential roles of the circRNA-coded proteins and the mechanism of their translation. Collectively, our data indicate that short IRES-like elements can drive extensive circRNA translation, which may represent a general function of circRNAs.

## Result
**Unbiased identification of short IRES-like elements.** Previously we reported that GFP-coding circRNAs can be translated from different viral or endogenous IRESs[12,13]. Similar circRNA translation reporters were also used by others[21], and were also previously validated using an in vitro translation assay[22]. In addition, we further confirmed that the GFP proteins were mainly translated from the circRNAs rather than the linear splicing byproducts generated from transcription readthrough or pre-mRNA trans-splicing[12].

Surprisingly, we found that three of the four short poly-N sequences designed as negative controls for IRESs could also promote GFP translation from circRNAs (Supplementary Fig. 1a). This observation indicates that certain short elements other than known IRESs are sufficient to initiate circRNA translation. To systematically identify additional sequences that drive circRNA translation, we adopted an unbiased screen approach originally developed to identify splicing regulatory

cis-elements[23–25]. Briefly, a library of random 10-nt sequences was inserted before the start codon of circRNA-coded GFP[26], which was transfected into 293T cells to generate circRNAs that can be translated into intact GFP (Methods and Supplementary Data 1). The green cells with active circRNA translation can be recovered with fluorescence-activated cell sorting (FACS), and we used RT-PCR with junction-specific primers to amplify the inserted sequences from circRNAs. The inserted decamers can be subsequently sequenced to identify the IRES-like elements that drive circRNA translation (Fig. 1a).

To achieve full coverage of entire library, >100 million cells were transfected (Fig. 1a). We sequenced the inserted fragments from both dark and green cells using high-throughput sequencing, and extracted the hexamers in the green and dark cells (Fig. 1b and Supplementary Fig. 1b) to identify 97 hexamers significantly enriched in green cells (Supplementary Data 2). These enriched sequences are generally A- and U-rich despite the pre-sorting library has roughly even base composition (Fig. 1c and Supplementary Fig. 1c), and have strong dinucleotide biases toward AC, AG, AT, and GA (Fig. 1d). Based on sequence similarity, the 97 enriched hexamers were further clustered into 11 groups to produce consensus motifs (Fig. 1e, top). Most consensus motifs are AU-rich motifs and enriched in 3′-UTR of linear mRNA (Supplementary Fig. 1d). Consistent with the previous report that m6A modification can drive circRNA translation[13], several enriched hexamers contain the RRACH signature for the m6A modification; however, they were not prevalent enough to be clustered into a consensus motif.

The IRES-like activities of each cluster were further validated by inserting the representative hexamers (or the depleted control hexamers) into the circRNA reporter to examine circRNA translation (Fig. 1e, bottom). All the reporters containing the enriched hexamers showed robust GFP translation from circRNAs, whereas the GFP productions from control hexamers were barely detectable (Fig. 1f). Furthermore, the enriched hexamers showed comparable activities as two short m6A-containing sequences that were previously reported to drive circRNA translation (Fig. 1g)[13]. In addition, the circRNA levels were similar in all reporters containing different sequences as judged by RT-PCR and northern blot (Fig. 1e–g and Supplementary Fig. 1e), suggesting the differences in GFP production are due to their distinct activities in driving translation rather than in affecting back-splicing.

Our circRNA reporters contain a 3-nt ATC sequence between the start codon and the library (Fig. 1h). To examine if this ATC may affect activities of the enriched hexamers, we deleted the ATC trinucleotide and found that all enriched hexamers tested can still direct circRNA translation from the mutated reporters, whereas no translation was observed from the depleted hexamers (Fig. 1h), suggesting that these enriched hexamers can independently drive circRNA translation.

We further examined the activity of enriched hexamers using in vitro synthesized circRNA that were further purified using HPLC[27] (Supplementary Fig. 1f). Compared to the controls, all tested circRNAs with candidate IRESs can be translated into luciferase in two cell lines (Supplementary Fig. 1g), and our newly identified elements have similar activity as known endogenous IRESs (5'UTR from Hsp70, CCGGCGG from Gtx, and UACUCCC from OR4F17)[28,29]. However, the viral IRESs from EMCV and CSFV showed a higher activity, probably because their activities are less dependent on the nuclear RNA binding proteins or modifications of the in vitro synthesized RNAs[30,31]. Collectively, these results indicated that our screen can reliably identify short sequences that drive circRNA translation, and thus we refer this set of 97 hexamers as IRES-like hexamers.

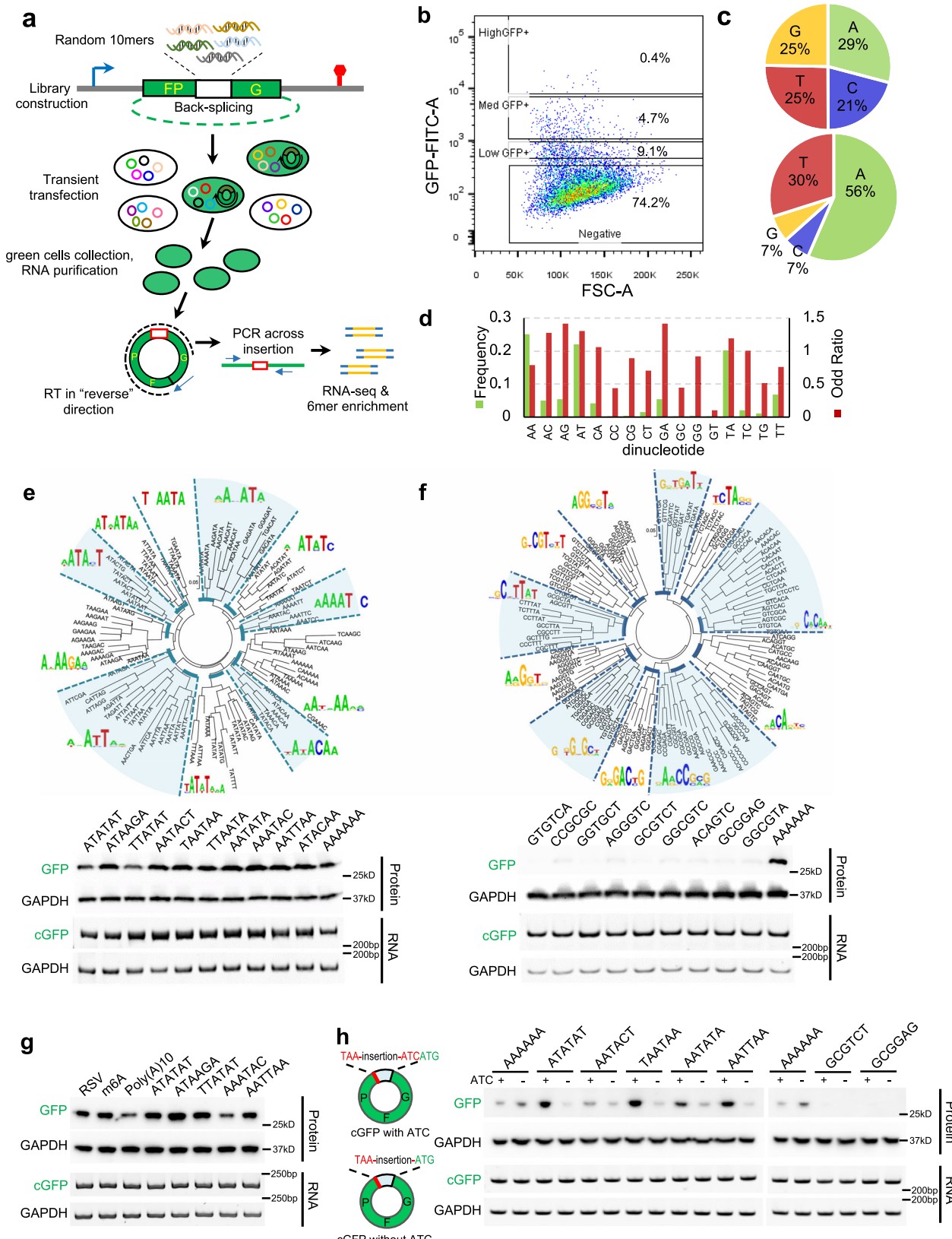

**IRES-like hexamers are enriched in endogenous circRNAs to drive translation.** We further compared the average frequency of the IRES-like hexamers *vs.* the control set of hexamers in different types of RNAs. In the linear mRNAs (all annotated mRNAs from RefSeq), the average frequencies of the IRES-like hexamers were similar to random hexamers or the hexamers depleted in green cells (Fig. 2a, left panel). Surprisingly, the average frequencies of IRES-like hexamers were significantly higher in all tested circRNA datasets[4,7,32,33] compared to the control hexamer sets (Fig. 2a), indicating that endogenous circRNAs are enriched with

**Fig. 1 Extensive IRES-like elements can drive circRNA translation.** Source data are provided in a Source Data file for panels (**e**–**h**). **a** Random decamers were inserted into pcircGFP-BsmBI, and the resulting library was transfected into 293T cells and the green cells sorted by FACS. The inserted sequences were recovered with RT-PCR and sequencing. The primers for RNA-seq library production were indicated by blue arrows. The enriched hexamers were identified computationally. **b** Flow-cytometry of cells transfected with circRNA reporter containing the 10-mer library. The cells were classified into four groups based on their GFP fluorescence (GFP negative, low, medium, or high GFP cells). The cells with medium and high fluorescence were sorted as "green cells". **c** Single nucleotide frequency in the starting library (top) and the sequences enriched in green cells (bottom). **d** Frequencies and odd ratios of the dinucleotide in the sequences enriched in green cells. The odd ratio is defined as the probability of a dinucleotide divided by the product of the probabilities of each base. **e** The 97 enriched hexamers (i.e., IRES-like elements, z score >7) were clustered into 11 groups with the consensus motifs shown as pictogram (top). Representative hexamers in each cluster were inserted back into the circRNA reporter, which were transiently transfected into 293T cells. The GFP was assayed by western blot at 48 h after transfection (bottom). **f** The 122 depleted hexamers (i.e., negative control, z score < -7) were clustered into 11 groups and the consensus motifs were shown as pictogram (top). Representative hexamers in all clusters were tested using the same condition as panel (**e**), with AAAAAA as the positive control. **g** Comparison of newly identified IRES-like elements with m6A sites (RSV and m6A) for the activity to drive circRNA translation using the same condition as panel (**e**). **h** Effects of neighboring sequence on circRNA translation. The enriched and depleted hexamers were inserted into circRNA reporters with or without ATC trimer (partially resemble Kozak sequence). The samples were analyzed using the same condition as panel (**e**). The circRNA reporters inserted with poly-A sequence were loaded twice as control.

short IRES-like elements. Since these IRES-like hexamers were independently identified from unbiased screen of random sequences, such enrichment strongly suggests that circRNAs may be positively selected for their ability to be translated.

The 97 IRES-like hexamers account for ~2% of the entire hexamer population ($4^6 = 4096$), indicating that many sequences may contain an IRES-like hexamer by chance. In another word, there would be one IRES-like element in about every 50 hexamers (97/4096). Since a sequence of length N can be broken into N-5 consecutive hexamers, and >99% circRNAs are longer than 100-nt[20] (Supplementary Fig. 2a), most circRNAs should contain internal IRES-like short elements by chance. Therefore, almost all open reading frames (ORFs) in circRNAs could potentially be translated from IRES-like short elements, implying most circRNAs may not require an additional known IRES to drive its translation.

To directly test this surprising hypothesis, we generated a series of circRNA reporters containing the coding sequence of GFP (717nt ORF, with 4 IRES-like elements) but without canonical IRES at the upstream (Fig. 2b). We confirmed the efficient circRNA expression in all the RNA samples using northern blot with the optional RNase R treatment (Supplementary Fig. 2b), and measured the GFP production with western blots. The experiments were conducted in both 293T cells (Fig. 2b) and in cells lacking a large T antigen (Supplementary Fig. 2c). The control reporter containing the m6A site was reliably translated[13], and deletion of stop codon led to production of GFP concatemers through the rolling circle translation of circRNA (Fig. 2b). When deleting all untranslated sequences between the start and stop codon in the circRNA, the intact GFP translation is abolished, presumably because there is no room for any sequences to function as the IRES. However, when deleting the stop codon, the circRNA containing only the GFP-coding sequence can also be translated in a rolling circle fashion to produce GFP concatemers, presumably through an internal sequence function as an IRES (Fig. 2b, Supplementary Fig. 2c). Interestingly, the rolling circle translation can produce some huge GFP concatemers (the huge proteins could not be efficiently transferred to the membrane and thus may be underestimated). To eliminate the possible artifacts from rolling circle transcription of plasmid, these reporter plasmids were stably inserted into genome using Flp-In system, and the similar rolling circle translations were observed in these stably transfected cells (Supplementary Fig. 2d).

The rolling circle translation initiated from the internal coding sequence is not limited to the GFP gene, as the circRNAs containing only the ORF sequences of two different luciferase genes can also be translated from the internal coding sequence (Fig. 2c, Supplementary Fig. 2e–g). Interestingly, the rolling circle translation from circRNAs apparently produced more proteins

than the circRNAs with stop codons (Fig. 2b and Supplementary Fig. 2c), suggesting that initiation of circRNA translation may be the rate-limiting step as the ribosome recycling and reinitiation is unnecessary for rolling circle translation[22,34,35].

**Trans-acting factors that bind to IRES-like short elements to promote circRNA translation.** We next seek to determine the molecular mechanisms by which these IRES-like short elements initiate cap-independent circRNA translation. An earlier report showed that short sequences may function as IRES by pairing with certain regions of 18S rRNA (i.e. active region)[19]. However, we found little correlation between our newly identified IRES-like elements to these "active 18S rRNA regions" (Supplementary Fig. 3a), suggesting these elements may not function by paring with 18S rRNA.

Previously we found that m6A reader YTHDF3 can recognize N6-methyladenosine in circRNA to directly recruit translation initiation factors[13]. By analogy we hypothesized that the IRES-like short elements may also function as regulatory cis-elements to recruit trans-acting factors that promote translation[36]. To identify such trans-acting factors, we used consensus sequences of IRES-like elements as bait for affinity purification of specific binding proteins. The chemically synthesized 20-nt RNA oligonucleotides containing three copies of IRES-like hexamers and a 5′ end biotin modification were incubated with HeLa cell extracts, and the RNA-protein complexes were purified with streptavidin beads[23–25] (Fig. 3a). We found that all the five RNA probes consist of IRES-like elements showed robust binding of several proteins, whereas the negative sequence (ACCGCG) had a weak background of non-specific RBP binding (Fig. 3b). The specific protein bands in each lane were subsequently analyzed with mass spectrometry (LC-MS/MS), and the top candidates were identified as candidate trans-acting factors (Fig. 3b).

In total 58 protein candidates were identified with high confidence, with many overlapping proteins in different baits (Supplementary Data 3). The majority of these proteins are known to bind RNAs and can be clustered based on the protein–protein interaction into two major groups, which are involved in RNA processing (e.g., hnRNPs) or in mRNA translation (e.g., the ribosomal proteins) (Fig. 3c). The identified RBPs are enriched for regulatory function in RNA processing, splicing, transport and stabilization as judged by gene ontology (Supplementary Fig. 3b).

To validate the function of these proteins, we fused the candidate RBPs to a programmable RNA binding domain (i.e., Puf domain) that can be designed to bind any 8-nt RNA sequences[37–40], and co-expressed the fusion proteins with the circRNAs containing their cognate targets. We chose to test

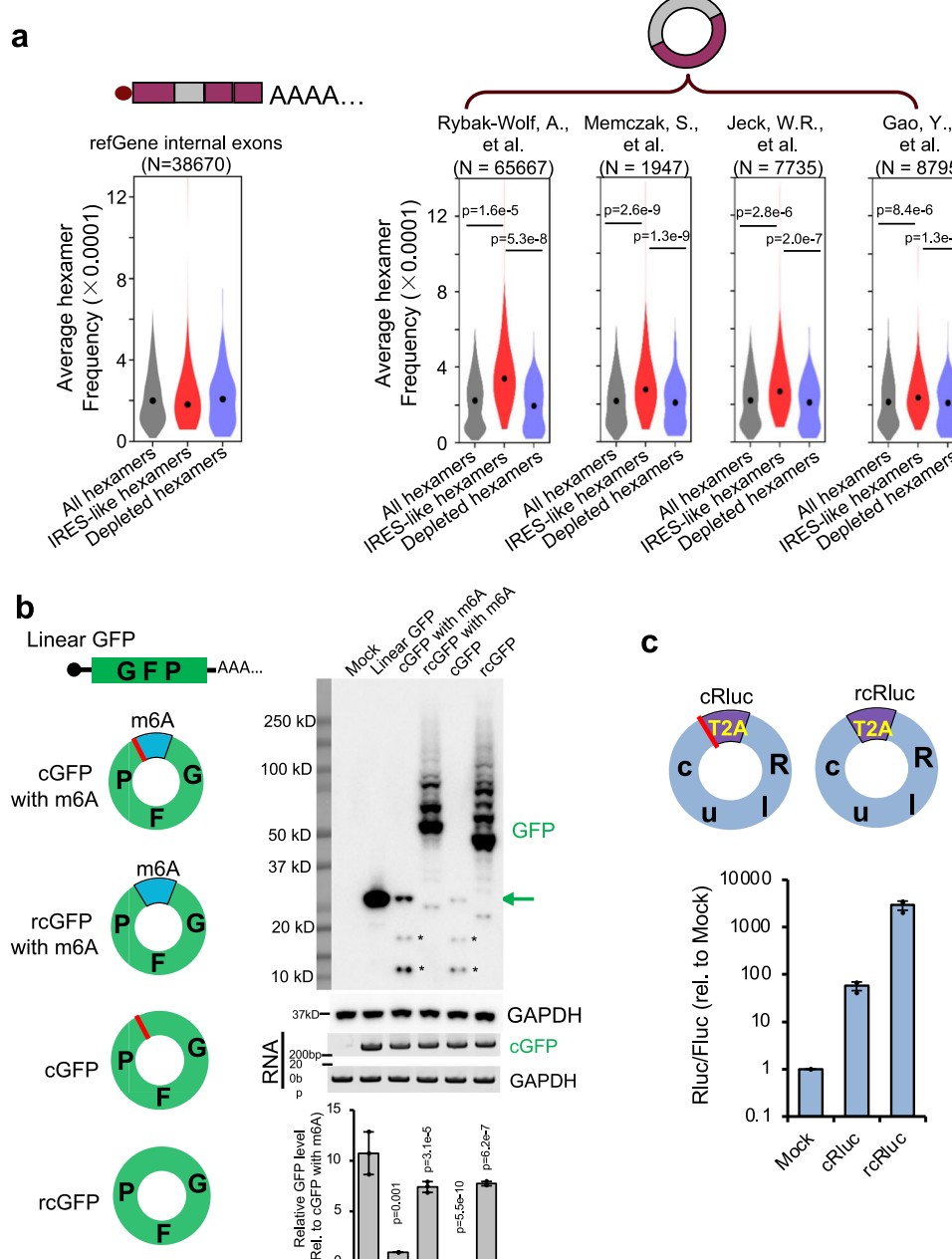

**Fig. 2 CircRNAs contain many IRES-like elements that initiate translation.** Source data are provided as a Source Data file for panels (**b**) and (**c**). **a** IRES-like elements are significantly enriched in circRNAs. Average frequencies of different types of hexamers (all hexamers, IRES-like hexamers and depleted hexamers) in the internal exons of linear mRNAs vs. circRNAs were plotted. N: the number of RNA sequences in each dataset. The p-values were calculated with two-sided Kolmogorov–Smirnov test. **b** Translation of circRNAs can be initiated by internal coding sequence. Left panel, schematic diagrams of expression reporters for a linear mRNA and four circRNAs that code for GFP. From top to bottom: linear GFP mRNA; circRNA with an m6A site at upstream of the start codon; circRNA with an upstream m6A site and no stop codon; circRNA with start codon immediately following the stop codon; circRNA containing only the coding sequence (no stop codon or UTR). Red line, stop codon. The circRNA plasmids were transfected into 293T cells, and samples were analyzed by western blot at 2 days after transfection (right panel). The level of GAPDH was measured as a loading control. The green arrow indicates the full-length GFP, and the asterisks indicate the truncated GFP proteins translated using internal start codons. The bar graph represents the quantification of GFP protein levels relative to GAPDH. The protein levels were also normalized to the RNA. **c** Translation of the circRNA-coded Renilla luciferase (Rluc) using internal IRES-like elements. Top: schematic diagram of two Rluc circRNAs, where the Rluc ORF was split into two parts so that the full-length Rluc ORF can only be generated through back-splicing. The cRluc contains a sequence coding a T2A peptide by which the translation product can be cleaved into full length Rluc protein. rcRluc does not contain stop codon. Bottom: dual luciferase assay of cRluc and rcRluc. Control and circular Rluc plasmids were co-transfected with Fluc (firefly luciferase) reference reporter into 293T cells. The cells were lysed at 48 h after transfection for luminescence measurement using luminescence reader, and the relative luminescence signals were plotted.

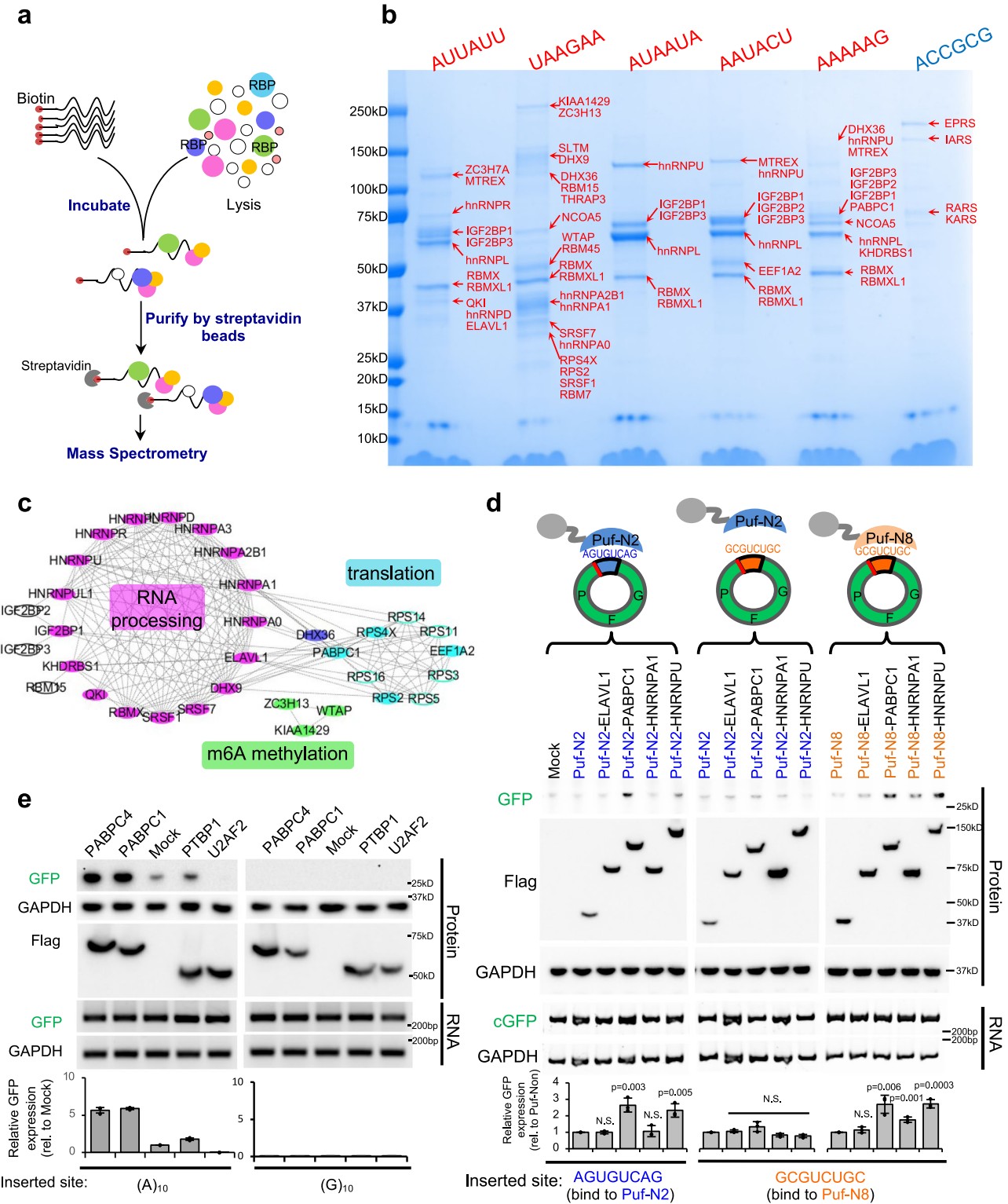

several factors that are known to specifically bind AU-rich elements (ELAVL1, PABPC1, hnRNP A1, and hnRNP U), but excluded the factors that bind all RNA baits with low sequence specificity (e.g., RBMXL1 and RBMX). We found that the specific tethering of PABPC1 and hnRNP U clearly promoted translation of the circRNA, whereas the ELAVL1 (HuR) and hnRNP A1 showed no activity when tethered to the same site (Fig. 3d). Such translation-promoting activity required the specific binding of *trans*-acting factors to circRNAs, as disrupting the Puf-RNA

interactions can abolish the regulatory effect and restoring the specific interaction can rescue the activity (Fig. 3d).

PABPC1 is an abundant protein that binds poly-A or AU-rich sequences[41]. We further examined the role of PABPC1 in regulating circRNA using reporters contain a short IRES-like element (poly-A) (Fig. 3e, left panel). We found that over-expression of PABPC1 and its paralogue PABPC4 can indeed promote translation of circRNA-coded GFP, whereas the circRNA translation was slightly affected by another polypyrimidine binding

**Fig. 3 Systematically identification of *trans*-factors that recognize IRES-like elements.** Source data are provided as a Source Data file for panels (**d**) and (**e**). **a** Schematic diagram of RNA affinity purification. Biotin-labeled RNAs containing consensus motifs of IRES-like elements were incubated with HeLa cell lysate, and RNA-protein complexes were purified by streptavidin beads. The proteins were further identified by mass spectrometry. **b** Identification of *trans*-factors bound by each RNA probe. The probes presenting five consensus motifs of IRES-like elements (red) and a control probe (blue) were used (full sequence in Supplementary Data 2). The total eluded proteins were separated with SDS-PAGE, and each band was cut and analyzed by mass spectrometry. The top three identified proteins in each band were labeled. **c** Protein–protein interaction network of identified *trans*-factors. Top proteins bound by all RNA probes (i.e., IRES-like elements) were analyzed by STRING and clustered into two main groups by MCODE tool. **d** Validating the activity of *trans*-factors. The circRNA reporter inserted with two depleted hexamers in tandem were co-transfected into 293T cells with different Puf-fusion proteins that specifically recognize an 8-nt target in the inserted sequences. The resulting cells were collected at 2 days after transfection to analyze by western blot and RT-PCR. Different pairs of Puf proteins and 8-nt targets were used as specificity control. Puf-N2 specifically binds AGUGUCAG, whereas Puf-N8 specifically binds to GCGUCUGC. The bar graph represents the quantification of GFP levels relative to GAPDH. The protein levels were also normalized to the RNA (N.S. not significant). **e** Validation of PABPC1 activity. The expression vector of PABPC1 and various control RBPs were co-transfected with circRNA translation reporter containing $(A)_{10}$ or $(G)_{10}$ sequences before the start codon, and the protein products were assayed at 48 h after transfection. The bar graph represents the quantification of GFP levels relative to GAPDH using the western blot. The protein levels were also normalized to the RNA, and the relative amount of GFP translation was calculated by dividing the mock transfection.

protein PTBP1 that was previously reported to enhance IRES activity. Interestingly, for unknown reason, the co-expression of U2AF2 seemed to inhibit circRNA translation. As a control, over-expression of these proteins showed no effect on the circRNA reporters containing the $(G)_{10}$ sequence (Fig. 3e, right panel). Furthermore, we used RNA-IP to examine the binding between PABPC1 with circRNAs. As expected, the PABPC1 indeed binds more tightly to the circRNAs containing IRES-like short elements compared to the control sequences (Supplementary Fig. 3c), supporting its role to promote circRNA translation by directly recruited to circRNAs through IRES-like elements. Taking together, our results showed that certain RBPs are capable to recognize these IRES-like elements to promote cap-independent translation of circRNAs, exemplifying a new model for circRNA translation initiation independent of canonical IRESs.

**Identification of endogenous circRNA-coded proteins**. We further examined the molecular characteristic of the putative circRNA-coded proteins. In accordance with previous observation[15], we found that the exons in the 5′ end of a pre-mRNA are more likely to be included in the circRNAs (Fig. 4a), suggesting that many circRNAs could potentially code for N-terminal truncated isoforms of their host genes. Consistently, 79% of circRNA exons are from the coding region, while 17% of circRNA exons spanning 5′-UTR and coding region and 4% spanning coding and 3′-UTR region (Fig. 4a inset).

Using the dataset of full-length human circRNAs[33], we examined the putative circRNA-coded ORFs that are longer than 20 amino acids. A large fraction of endogenous circRNAs (67%) can code for proteins overlapping with their host genes (i.e., translated in the same reading frame), including 14% of circRNAs that can be translated in a rolling circle fashion (named as overlapped cORF and rcORF, respectively, purple pie slices in the left panel of Fig. 4b). In addition, 16% of human circRNAs can code for proteins that are homologous to other known proteins (homologous cORF and rcORF, brown pie slices in Fig. 4b, left panel), whereas 10% circRNA-coded proteins are not homologous to any known proteins. Only 7% of circRNAs have no ORF longer than 20 aa. In comparison, a much larger fraction of circRNAs from two different controls (reversed or shuffled sequences) do not contain ORFs longer than 20 aa (Fig. 4b, right), and the remaining control circRNAs are more likely to code for proteins that are not homologous to any known proteins.

To systematically identify circRNA-coded proteins, we searched raw mass spectra from public datasets of tandem mass spectrometry (MS-MS) for possible peptides across the back-splice junctions of all circRNAs[20,33] (Fig. 4c). Two sets of high-resolution human proteomic data from 30 tissues and 6 cell lines

were selected[42,43]. Since a comprehensive dataset of circRNAs in different human tissues is lacking, we merged all circRNAs from different cell lines/tissue, which may increase the false positive rate. However, we used a series of stringent filters and statistic tests in the analysis of proteomic data to alleviate the false-positive discovery (see "Methods").

We identified 2721 mass spectra across 990 back-splice junctions from 646 human genes, all of which contain putative cORF or rcORF longer than 20 amino acids (Fig. 4d, Supplementary Data 4). These newly identified spectra were not previously assigned to known proteins or isoforms. Interestingly, the fraction of circRNAs with rolling circle translation increased as the additional filters were applied in our search (Fig. 4d). Such an increase is consistent with a higher translation efficiency when reinitiation is not required (Fig. 2b). Alternatively, the increased detection rate for rcORFs may partially due to an inherent bias of detecting the proteins containing multiple copies of the same peptides.

More than 80% of the identified circRNA-coded peptides overlapped with the translation products of their host genes (Fig. 4d), suggesting that circRNAs preferably produce different translation isoforms of the host genes. Interestingly, their host genes are significantly enriched with the functions in RNA translation, RNA splicing/processing, and platelet degranulation (Supplementary Fig. 4), implying that many circRNAs code for new protein isoforms with potential roles in regulating these processes. The enrichment in platelet degranulation may also provide a functional implication to the previous observation that the circRNAs are highly expressed in platelet[44,45].

For many circRNAs, we identified multiple spectra to support the same back-splice junctions, including 80 circRNAs with >10 different spectra across their back-splice junctions (Fig. 4e). The circRNA-coded peptides are mostly presented in a small set of cells or tissues, with 60-80% of circRNA-coded peptides being identified only from a single sample in both MS-MS datasets (Fig. 4f, blue bars). In comparison, the peptides encoded by the adjacent splicing junctions of linear mRNAs are more ubiquitously expressed, with some peptides being found in all samples (Fig. 4f, gray bars).

**circRNA-coded proteins have low abundance due to rapid degradation**. We next analyzed the mass spectra supporting circRNA-coded peptides and compared them to the known proteins encoded by canonical linear mRNAs. As expected, the numbers of newly identified circRNA-coded peptides were positively correlated with the numbers of total peptides in two independent datasets (Fig. 4g). In addition, applying additional fractionations in the same sample (e.g., 39, 46, and 70 fractions of HeLa cells) increased the numbers of circRNA-coded

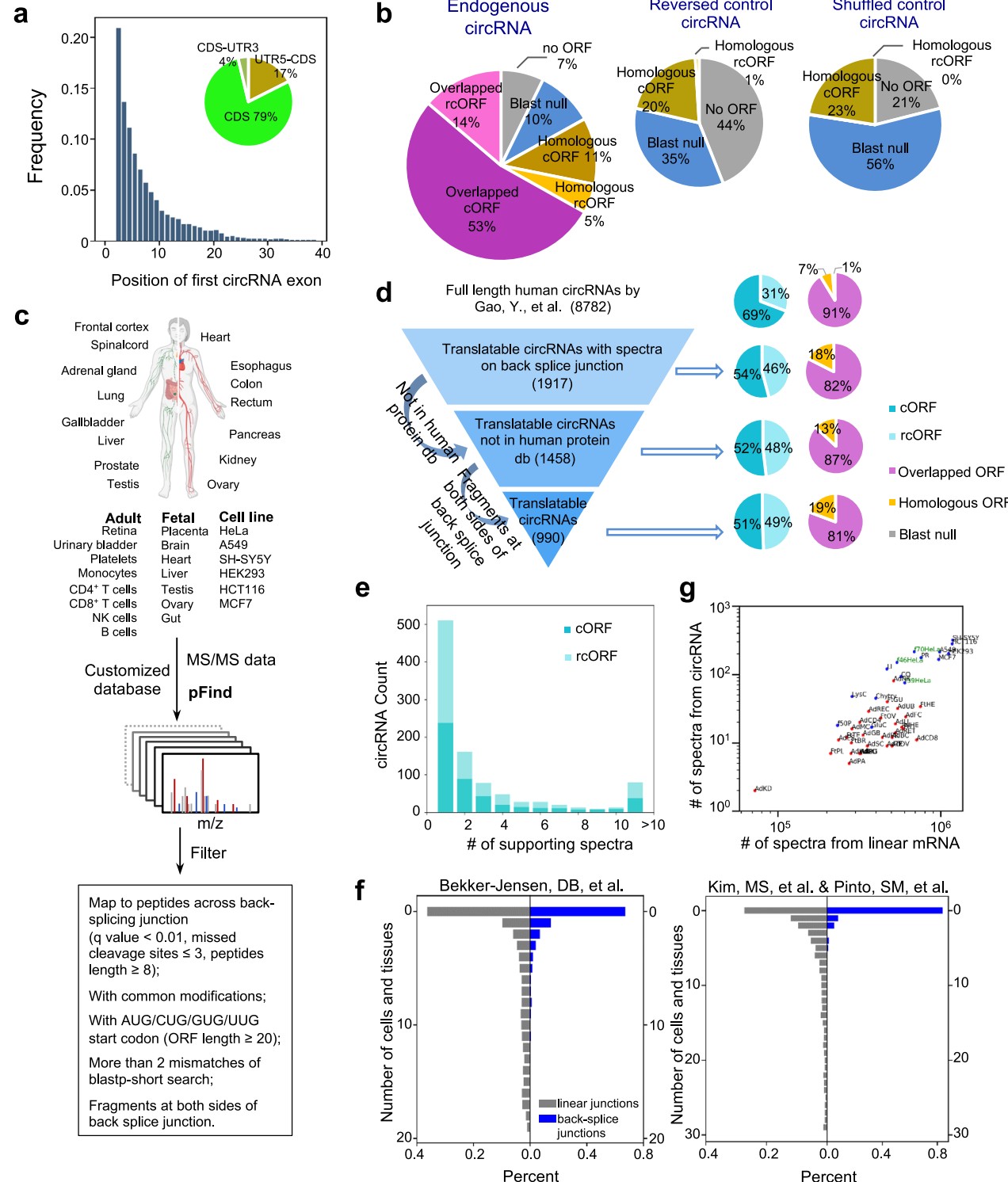

peptides (Fig. 4g), suggesting that the proteomic identifications of circRNA-coded proteins have not reached saturation and the spectrum number of supporting peptides is roughly correlated to the abundance of these "new proteins".

To estimate the relative abundance of circRNA-coded proteins, we compared the peptides across back-splice junction *vs.* those across adjacent splicing junctions of the host linear mRNA. The circRNA-coded peptides generally have a smaller number of supporting spectra as compared to the peptides encoded by linear adjacent splice junctions (Supplementary Fig. 5a), suggesting that

the circRNA-coded peptides generally have a lower abundance than their linear counterparts. Consistently, the *q* values of the mass spectra supporting circRNA-coded peptides were much higher than their linear counterparts (Supplementary Fig. 5a). Since the more abundant proteins are usually supported by MS data with higher confidence, this result again suggested that the circRNA-coded proteins generally have lower abundance than proteins encoded by linear mRNAs.

By analyzing the sequence composition across splicing junctions, we found that lysine and arginine are enriched at the

**Fig. 4 Identification of circRNA-coded proteins. a** Position distribution for the first circRNA exon in their host genes. Full-length circRNA sequences were analyzed based on previous datasets, and the histogram was plotted according to the exon number of the first circRNA exon in host genes. The inserted pie chart presents the percent of circRNAs overlapping with different mRNA regions. **b** Survey of potential circRNA-coded products. Left, percent of endogenous circRNAs with an ORF > 20 aa. Purple pie slices: circRNAs translated regularly (cORF, dark purple) or in a rolling circle fashion (rcORF, light purple) into proteins that are partially overlapped with their host genes for at least 7aa; brown slices: circRNAs translated in a regular fashion (cORF, dark brown) or a rolling circle fashion (rcORF, light brown) into proteins that are not overlapped with their host genes but are homologous to other known proteins; blue pie slices: circRNAs translated into proteins that are not homologous to any known protein (cORF and rcORF combined); gray slices: circRNAs do not contain any potential ORF longer than 20 aa. Right panels: the same analysis of putative coding products from two control circRNA sets: reversed sequences or randomly shuffled sequences of the endogenous circRNAs. **c** Schematic diagram to identify circRNA-coded proteins using proteomic datasets. **d** Left, computational filters sequentially applied to identify translatable circRNAs and the numbers of circRNAs passing each filter. Right, the percentage of different types of circRNA-coded ORFs in the circRNAs passing each filter. The definition of different circRNA-coded ORFs is the same as panel (**b**). **e** Distribution of the supporting spectra for each translatable circRNA. **f** Distribution of the number of cell lines and tissues for each translatable circRNA in two proteomic datasets. **g** Comparison of the numbers of spectra from linear mRNAs vs. circRNAs. Abbreviations of different tissues and cell lines are listed in Supplementary Data 4. The proteomic data from Bekker-Jensen, DB, et al (blue) and Kim, MS, et al. & Pinto, SM, et al (red) were used. Green words indicate the 39 fractions, 46 fractions and 70 fractions from HeLa cells using high-capacity offline HpH reversed-phase LC.

---

−1 position of all splicing junctions (Supplementary Fig. 5b). Specifically, >70% of splice junctions contain at least one lysine or arginine regardless of linear or back-splice junctions. Since the current proteomic samples are mostly lysed by trypsin with a cleavage site of lysine or arginine, such sequence bias leads to a significant depletion of peptides across splice junctions, which is consistent with an earlier report[46]. Unlike the proteins encoded by linear mRNAs, the circRNA-coded proteins can only be recognized by peptides across back-splice junctions, which could partially explain why the circRNA-coded peptides are difficult to identify. Further optimization of MS-MS using different proteases may identify additional circRNA-coded proteins.

In addition to these technical difficulties, multiple factors may also contribute to the low abundance of circRNA-coded proteins. First, the abundance of most circRNAs is generally lower than their linear counterparts[47–49], which should partially contribute to the low abundance of the circRNA-coded peptides. In addition, the low abundance of circRNA-coded proteins may also due to a slow protein synthesis and/or a fast protein degradation. It is well known that the efficiency of cap-independent translation is relatively low[50,51]; however, the stability of circRNA-coded proteins is still unclear. Although the majority of circRNA-coded protein sequences overlap with proteins encoded by the host genes, it is possible that the extra peptides specifically encoded by back-splicing junction may make the protein unstable.

To directly address this possibility, we selected three C-terminal peptides specifically encoded by the back-splice junctions of different circRNAs, and fused them to the C-terminus of GFP in the circGFP reporters (Supplementary Fig. 5c). Tethering the potential degron onto a reporter protein is a routine assay to measure protein degradation[52,53]. We found that the GFPs with different circRNA-coded tails are expressed in much lower levels compared to the one without these tails (Supplementary Fig. 5d). Moreover, the expression level of the fusion proteins with circRNA-coded tails increased upon the proteasome inhibition by MG132 treatment, whereas the GFP without such tails or with control peptide tails (N-terminal peptides of the same gene or a V5 epitope tag) were essentially unaffected (Supplementary Fig. 5e). In contrast, the levels of all GFP fusion proteins were not affected by chloroquine treatment that inhibits autophagic protein degradation (Supplementary Fig. 5e). These observations indicate that the circRNA-coded C-terminal tails can indeed destabilize many circRNA-coded proteins, and such degradation was mainly mediated by proteasomes.

**Translation products derived from circRNAs.** Although the endogenous circRNAs were found to contain more cORFs (with the cORF:rcORF ratio of ~3.4, Fig. 4b), further analyses showed

that ~50% of the circRNA-coded peptides are derived from the endogenous circRNAs containing rcORFs (Fig. 4d), suggesting rcORFs are more efficiently translated from circRNAs (consistent with observations in Figs. 2b, c). Since the endogenous circRNAs had not been reported to undergo rolling circle translation, we seek to further determine whether these identified circRNAs with rcORFs are indeed translated in different cells.

Detecting translation products of endogenous circRNAs is technically difficult because they differ from the canonical products only at the back-splice junctions. Generating new antibodies against the short fragments coded by the back-splice junctions is time consuming and sometimes unreliable, and thus we labeled the candidate circRNA products with an epitope tag. We selected three circRNAs (circPSAP, circPFAS, and circABHD12) with high-quality MS evidence (Fig. 5a and Supplementary Fig. 6a), and constructed back-splicing reporters to ectopically express these circRNAs in two different cell types[26]. An in-frame V5 epitope tag was inserted into the rcORFs to facilitate the detection of translation products (Fig. 5b). We found multiple protein bands in 293T and SH-SY5Y cells transfected with all circRNAs tested, suggesting that these circRNAs undergo robust rolling circle translation to produce protein concatemers (Fig. 5c and Supplementary Fig. 6b). Interestingly, the cells transfected with circPSAP produced a faint band of protein concatemer and a much stronger band corresponding to the product of a single cycle of circRNA translation, suggesting a low translation processivity for this circRNA.

Since the translation of rcORFs can generate protein concatemers with repeat sequences, they are likely to induce protein misfolding and aggregation that often leads to rapid protein degradation. To measure the stability of these proteins translated from rcORF, we treated the cells with short exposure of MG132 or chloroquine, and found that the translation products of two circRNAs (circPSAP and circPFAS) were increased by MG132 treatment but not by chloroquine while the control protein GFP was not affected (Fig. 5d). These findings suggested that some rolling circle translation products were rapidly degraded in cells through proteasome pathway, which are consistent with the observation that the circRNA-coded proteins are generally low abundant inside cells (Supplementary Fig. 5). Intriguingly, the translation product of circABHD12 seems to be relatively stable despite containing repetitive sequences.

We next examined whether the rolling circle translation of endogenous circRNAs is indeed driven by the newly identified IRES-like short elements. The circPFAS contains two IRES-like hexamers, and we made individual mutations on both hexamers to examine their effects on circPFAS translation (Fig. 5e, left). The mutation on one IRES-like hexamer (AAGAAG) dramatically

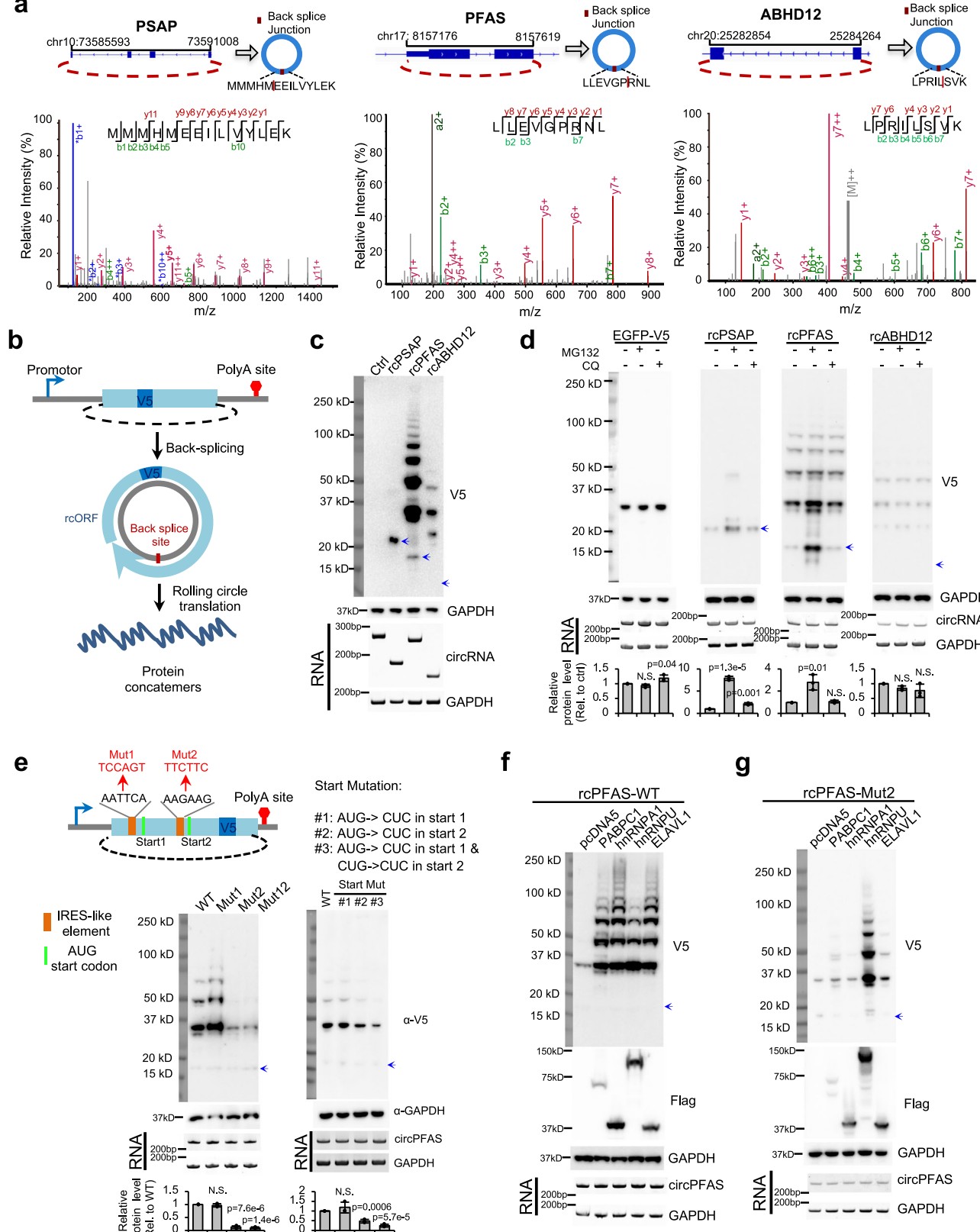

reduced the translation of circPFAS, whereas the mutation on the other element (AATTCA) had no effect, suggesting that the AAGAAG hexamer is the major translation initiation element in circPFAS (Fig. 5e and Supplementary Fig. 6c). Interestingly, a trace amount of translation product was observed even when mutating both IRES-like elements, suggesting that other

unknown *cis*-elements may also help to promote circPFAS translation. This observation is not a surprise since we used a strong cutoff (enrichment z score >7) in the identification of the 97 IRES-like hexamers, and thus may miss some elements with weak activity. We further mutated the downstream AUG start codons to confirm that the second AUG was the major

**Fig. 5 Rolling circle translation of endogenous circRNAs.** Source data are provided as a Source Data file for panels (**c–g**). **a** The higher-energy collisional dissociation MS/MS spectrum of the peptide across the back-splice junctions of the human circPSAP (MMMHMEEILVYLEK), circPFAS (LLEVGPRNL), and circABHD12 (LPRILSVK). The annotated b- and y-ions are marked in red and green, respectively. **b** The rcORF translation reporters. The coding region of the endogenous rcORF was inserted into a back-splicing reporter, with an in-frame V5 epitope for detection. **c** Three rcORF reporters were transfected into 293 cells, the cells were collected at 48 h after transfection and analyzed by western blotting and RT-PCR. The blue arrows represent the predicted MW of the single cycle of translation product from cPSAP (22.4kD), cPFAS (15.6kD), or cABHD12 (10.5kD). **d** The rcORF translation reporters were transfected into 293T cells and then treated with 10 μM MG132 for 2 h, or 10 μM chloroquine for 4 h before cell collection. The bar graph represents the quantification of protein levels relative to GAPDH, which were also normalized to the RNA (N.S. not significant). **e** The circPFAS contains two IRES-like hexamers (AATTCA and AAGAAG), which were mutated into neutral sequences (mut1 and mut2). The downstream AUG codons were mutated into CUC (Start Mut #1 and #2), and the non-canonical start codons CUG at downstream of the AAGAAG hexamers were further mutated into CUC (Start Mut #3). The effects of these mutations on protein production were determined with western blotting using similar procedure as panel **c**, with relative protein changes represented by bar graphs (N.S. not significant). **f** The back-splicing reporter of rcPFAS was co-transfected into 293T cells with the expression vectors of various *trans*-acting factors that bind to the newly identified IRES-like elements. The cells were collected and analyzed using same procedures as panel (**c**). **g** The circRNA with mutated IRES-like hexamer AAGAAG (Mut2) was co-expressed with the same set of *trans*-acting factors, and the production of rolling circle translation were measured using same experimental conditions described in panel (**f**).

translation start site of circPFAS, as its mutation significantly reduced the translation efficiency (Fig. 5e, right). Interestingly, the circRNA translation is reduced but not completely abolished with the AUG mutation, and additional mutations of nearby in-frame CUG site can further reduce translation (Fig. 5e). This result suggests that the cap-independent translation driven by IRES-like hexamers can also use non-canonical start codon (e.g., CUG), which is consistent with previous reports[54].

We next examined if the rolling circle translation of endogenous circPFAS can also be affected by the *trans*-acting factors that promote translation of circRNA reporters (Figs. 3d, e). We co-expressed the circPFAS with PABPC1, hnRNP A1, hnRNP U, or ELAVL1 in two different cell types (293T and SH-SY5Y cells), and found that all tested *trans*-acting factors significantly increased the rolling circle translation products from circPFAS (Fig. 5f and Supplementary Fig. 6d). This observation suggests that the *trans*-acting factors identified earlier (Fig. 3) may also promote the rolling circle translation of endogenous circRNA, and that the translation products of a single circRNA may be affected by multiple *trans*-factors. In addition, the mutation of the cognate IRES-like element (AAGAAG) reduced the translation enhancing activities of PABPC1, hnRNP A1 and ELAVL1 on circPFAS, whereas the effect of hnRNP U is independent of this IRES-like element (Fig. 5g and Supplementary Fig. 6e). This difference is probably because these factors bind to the IRES-like hexamer with distinct specificities and affinities, or due to subtle differences in the mechanism by which various factors affect circRNA translation.

## Discussion
Several recent studies indicated that circRNAs can function as template to direct protein synthesis; however, the nature of circRNA translation is still under debate because other studies failed to detect the significant association of circRNA with polysomes[4,55,56]. In addition, the mechanism of circRNA translation is not clear. While the cap-independent translation of circRNA generally requires a viral or endogenous IRES, we have demonstrated that a short element containing m6A modification is sufficient to drive circRNA translation[13]. Here we surprisingly found that the requirement of IRESs can be easily fulfilled, and many short sequences (~2% of all hexamers) are capable of driving cap-independent translation of circRNAs. This finding is also consistent with an earlier report that in vitro synthesized circRNAs with mutated Kozak sequences can still be translated[22]. As a result, thousands of cytoplasmic circRNAs can be potentially translated, among which hundreds of circRNA-coded peptides were supported by mass spectrometry evidences. We further identified many RNA binding proteins that specifically recognize

these IRES-like short elements and function as *trans*-factor to promote cap-independent translation, providing a new paradigm on the mechanism of circRNA translation.

Many identified IRES-like short elements are AU-rich sequences that resemble a canonical polyadenylation signal (AAUAAA), raising a concern that they may cause circRNA cleavage followed by subsequent re-caping and polyadenylation to produce a linear mRNA of entire GFP. This concern was alleviated by the design of our screen. To recover the enriched sequence from green cells, we purified the total RNA and used RT-PCR to amplify across the insertion region with specific primers (Fig. 1a). Therefore, if the inserted sequences cause the cleavage and poly-adenylation, they would not be recovered by RT-PCR and thus would be depleted (rather than enriched) in the screen. Consistently, we cannot capture any free 5'-end near GFP starting codon using 5'-RACE. In addition, the canonical poly-A signal should cause RNA cleavage at ~20 nt downstream of AAUAAA motif, which would be after the initiation ATG codon and thus will destroy the ORF of a functional GFP. Collectively, the concern of the full-length translation products derived from cleaved circRNA with 5'-cap and 3'-polyA tail is unlikely.

We also examined the IRES activity using a traditional bicistronic reporter with two tandem ORFs encoding firefly and Renilla luciferases[57]. Both the known short IRESs from endogenous genes (CCGGCGG from Gtx, and UACUCCC from OR4F17) and the short IRES-like elements tested have little, if any, activity in direct cap-independent translation in a bicistronic vector (Supplementary Fig. 7a), although they all worked well in circRNA reporters (Supplementary Fig. 7b). The only exception is the long IRES sequence from Hsp70 that functioned in both reporters. A possible reason for the low sensitivity of the bicistronic reporter is that the linear mRNAs have higher translation background for empty control than circRNAs (~400 fold higher, Supplementary Fig. 7c), thus reducing its sensitivity of detection. Consistently, canonical bicistronic vectors were previously reported to produce high background due to various artifacts, including cryptic promoters, cryptic splicing sites, ribosome shunting, or reinitiation[16–18,58]. As a result, many data using such a system have been challenged[17,18]. Since circRNAs have no free 5' or 3' ends, they may be a more reliable system for the future test of IRES activity.

The majority of the identified translatable circRNAs code for new isoforms overlapping with canonical gene products, 50% of which can produce protein concatemers through a rolling circle translation. This finding raises an interesting open question regarding their biological functions. Because these circRNA-coded isoforms have a large overlap with the canonical host genes, the circRNA-coded isoforms may function as competitive

regulator of canonical isoforms or play a similar function in different subcellular location. The protein concatemers translated from circRNAs may also function as scaffold for assembly of large complexes, or form protein aggregations that are toxic to cells.

Compared to linear mRNAs, a relatively small number of circRNAs are reported to be associated with polysomes[55,56]. In addition, the minimal requirement for the cap-independent translation of circRNAs and the difficulties to detect circRNA-coded products presented an intriguing paradox. While there is no clear explanation, several factors may help to reconcile such contradiction. First, although most circRNAs have the capacity to be translated, it does not necessarily mean that they are indeed translated efficiently in vivo to produce stable proteins. Some products from pervasive translation of circRNAs may not be folded correctly and thus be rapidly degraded. This scenario is conceptually similar to the pervasive transcription occurred in many genomic regions on both directions, where most of the transcribed products are degraded and only a small fraction of products are stable and functional[59,60]. In support of this notion, we found that the short C-terminal tails specifically encoded by circRNA sequences across the back-splice sites can cause rapid protein degradations (Supplementary Fig. 5d). In another word, the protein folding and stability may function as the quality control step for circRNA translation.

The other possibility is that the initiation of cap-independent translation of circRNAs is less efficient than linear mRNAs, but the translation elongation rate should be comparable between circular vs. linear RNAs. As a result, the translation of most circRNAs may be carried out by monoribosomes rather than by polysomes, which could explain the lack of circRNAs in polysome-associated RNAs. Consistently, rolling circle translation of the reporter circRNA has produced much more products than the same circRNA containing stop codon (Figs. 2b and 2c), the latter of which requires reinitiation at each round of translation. In addition, we observed a larger fraction of circRNA-coded peptides from rolling circle translation in the analyses of mass spectrometry dataset (Fig. 4d vs Fig. 4b), suggesting that products of rolling circle translation are relatively more abundant.

We have, for the first time, identified the rolling circle translation products of several cellular circRNAs and validated them using plasmid expression system (Fig. 5). Because such proteins contain concatemeric repeats, they are probably misfolded to form protein aggregates with pathogenic properties. On the other hand, the misfolded protein products often induce unfolded protein response that leads to protein degradation and apoptosis. We think the biological functions of the circRNAs containing rcORFs will be an important subject for future study.

It is well accepted that the translation through cap-independent pathways is less efficient under normal physiological condition. However, under certain cellular stress condition (like heat stress) or in certain cell types (like cancer cells), the canonical cap-dependent translation is inhibited and the cap-independent translation may become more predominant[61,62]. Consistently we found that the translation of GFP from circRNA is promoted in heat shock conditions[13], suggesting that circRNA-coded proteins may be induced under such conditions (or in certain cells where canonical translation is suppressed), implying potential roles for the circRNA-coded proteins in stress response and cancer cell progression.

The finding that many short IRES-like elements can drive pervasive circRNA translation also has profound implications: many internal ORFs in linear mRNAs could be translated in a cap-independent fashion through the abundant IRES-like elements, especially when the cap-dependent translation is inhibited under stress conditions. In consistent with this hypothesis, several studies reported that various "alternative ORFs" or "non-coding regions"

may be translated into small peptides as judged by ribosome profiling or improved analyses of mass spectrometry data[63–66]. Such cap-independent "alternative translation" may serve as a new mechanism similar to alternative splicing in increasing human proteome complexity[50]. We speculate that some products of non-canonical translation may not have a specific function since they may be degraded rapidly. Analogous to the "pervasive transcription" of the genome, the abundant short IRES-like elements probably mediate "pervasive translation" of the transcriptome to generate the "hidden" proteome[67]. It will be intriguing to detect such 'dark matter' of proteome by systematically mining the available MS data, which might be technically difficult (but biologically important) because the levels of these non-standard peptides are generally lower than the standard proteins.

## Methods

**Plasmid library construction and screening**. In order to screen short elements for the initiation of circRNA translation, the previously described pcircGFP reporter[12] was modified to minimize the sequences between stop and start codon of circular GFP. Two BsmBI restriction sites were inserted between stop and start codons of GFP in pcircGFP reporter, called pcircGFP-BsmBI.

To produce the random 10 mer sequence library, we extended the foldback primer (ATTCCGTCTCAAGTAA(N10)ATCATGGAGACGCACTGTTTTTTTTCAGTGCGTCTCCATGA) with Klenow fragment (NEB), cut the resulting DNA with BsmBI and ligated into BsmBI digested pcircGFP-BsmBI. The ligation product was transformed into ElectroMax DH-5α (Invitrogen), and we obtained sufficient numbers of E. coli clones to achieve ~2-fold coverage of all possible DNA decamers (total 2 million clones). The resulting library was extracted through QIAGEN Plasmid Mega Kit, and transfected into 293T cells (20 μg/ per 15 cm dish) by using lipofectamine 2000 (Invitrogen). To cover the entire decamer space, totally 10 × 15 cm dishes were used.

48 h after transfection, cells were collected for FACS sorting using BD FACSAria II. To select the singlets, SSC-A vs FSC-A was used to select 293T cells (excluding very small and very large particles). Two round selections of singlets were used by SSC-W vs FSC-H and FSC-W vs FSC-H. Then FITC-A vs FSC-A were used to select GFP-positive cells. The circRNA reporters inserted with short poly-A and poly-G sequences were used as the positive and negative controls of the screen. In total 122 million cells were sorted, we collected 4 million cells without GFP fluorescence (negative controls), 13 million cells with low GFP fluorescence, 5 million cells with medium GFP fluorescence and 0.5 million cells with high GFP. Then RNAs were extracted, and sequencing library was generated by RT-PCR. RNA-seq was performed with Hiseq 2500.

**Cell cultures and Transfection**. 293T human embryonic kidney cell line, SH-SY5Y neuroblastoma cell line, and HeLa epithelial cell line were cultured in DMEM (high glucose) medium containing 10% fetal bovine serum (FBS, Hyclone). To transient transfect plasmids into cells, 2 μg of mini-gene reporters were transfected into cells in 6 well plate, using lipofectamine 3000 (Invitrogen) according to the manufacturer's instruction. After 48 h, cells were collected for further analysis of RNA and protein levels. To transfect circRNAs into cells, 200 ng of RNAs were transfected into cells in 24-well plate, using lipofectamine 3000 (Invitrogen) according to the manufacturer's instruction.

**Identification of enriched motifs in IRES-like elements**. More than 20 million reads were obtained from cells with different levels of GFP fluorescence, generating 10-nt sequences from each cell fraction. The inserted 10-nt sequences were recovered when the flanking sequences have exact match to the mini-gene reporter (24-nt upstream sequences and 30-nt downstream sequences were used for comparison). We used a statistic enrichment analysis to extract enriched motifs from decamer sequences recovered in green cells. Briefly, each inserted 10-mer was extended into a 14-mer by appending 2-nt of the vector sequence at each end to allow for cases in which IRES activity derived from sequences overlapping the vector. The resulting 14-nt sequences were broken into overlapping hexamers and the occurrences of different hexamers were counted from sequences recovered in cells with mid-high fluorescence vs. no fluorescence. The enrichment score of each 6-mer between two datasets was calculated using Z-test. Hexamers with score larger than 7 are defined as IRES-like elements, while less than −7 are defined as depleted motifs. The resulting elements were aligned with CLUSTALW2[68] to generate consensus motifs, which were plotted by Weblogo3[69].

**circRNA datasets**. Four published circRNA datasets (Fig. 2a) were retrieved from circBase (http://circbase.org/). The full-length circRNA dataset was created using the published ribominus RNA-seq data that was generated from the RNase R treated RNAs of HeLa cells (BioProject database of Genbank, accession number PRJNA266072). The circRNAs were detected by CIRI2 v2.0.6 package[70] with annotation of GENCODE v27[71]. Subsequently, the fraction of full-length circRNAs

without introns were selected, and their sequences were converted into bed format file for downstream analyses. The full-length sequences of these circRNAs were obtained using bedtools getfasta module.

**Semi-quantitative RT-PCR and real-time PCR**. Total RNAs were isolated from transfected cells with TRIzol reagent (Invitrogen) according to the manufacturer's instructions. Total RNAs (1 μg) were treated by gDNA eraser to remove genomic DNA, then reverse-transcribed with PrimeScript RT Reagent kit with gDNA Eraser (TaKaRa) using a mixture of oligo(dT) and random hexamer primers (following the manufacturer's instructions). Then, RT products (1 μl) were used as the template for PCR amplification (25 cycles of amplification). To reduce potential inconsistency from repeat exposure of different gels, we often mixed the samples of two different RT-PCR reactions (with GAPDH and GFP specific primers) together and run them on the same gel (the two products are in different sizes so that they can be clearly separated). The PCR products were separated on 10% poly-acrylamide gel electrophoresis (PAGE) gels, stained by SYBR Green I (Thermo Scientific) and scanned using ChemiDoc Touch Image system (BioRad).

**RNase R treatment**. To verify the expression of circular RNA, total RNAs were treated by RNase R. Total RNAs were heated at 95 °C for 5 min to denature RNA (RNAs may be partially fragmented during heat treatment), and then immediately placed on ice for 2 min, subsequently 10 μg of denatured RNAs were treated with RNase R (10 U) (epicentral) at 37 °C for 15 min and 42 °C for 15 min. Then samples were analyzed by northern blot or stored at −80 °C.

**Northern blot**. RNA samples were mixed with 2X RNA loading buffer, and heated at 65 °C for 5 min. Samples were separated by 1% formaldehyde agarose gel, and then transferred to nylon membrane (Millipore) at 15 V for 60 min by using Trans-Blot® SD Semi-Dry Transfer Cell (BioRad). Then RNAs were UV-crosslinked with nylon membrane by UV crosslinker (Analytik Jena) at an energy of 120,000 microjoules. DIG-labeled RNA probes (probe sequences in Supplementary Data 1) were generated through in vitro transcription, and hybrid to target RNA at 68 °C over night. The blots were visualized with DIG northern starter kit (Roche) according to the manufacturer's instructions.

**Western blot**. Cells were lysed in RIPA buffer with protease inhibitor cocktail (Roche), and the total cell lysates were resolved with 4-20% ExpressPlus™ PAGE Gel (GeneScript). The following antibodies were used: GFP antibody (Clontech: 632381) is diluted by 1:2000; V5 antibody (CST: 13202S) is diluted by 1:2000; Flag antibody (Sigma: F1804-1MG) is diluted by 1:2000; GAPDH antibody (Proteintech: HRP-60004) is diluted by 1:10000. The HRP-linked secondary antibodies (CST: 7076S) were used by 1:2000 dilution and the blots were visualized with ECL reagents (Bio-Rad).

**Dual Luciferase assay**. Reporters were transfected into 293T cells in 24-well plate (200 ng circular reporter and 50 ng reference reporter per well). At 48 h after transfection, cells were collected and lysed in passive lysis buffer. Dual-luciferase reporter assay system (Promega) was used to generate the luminescent signal (following the manufacturer's instructions), and the luminescence was measured by Bio-Tek synergy H1.

**Identification of *trans*-factors with RNA affinity purification**. The RNA affinity purification method was adopted from the previously described protocol[23]. For each biotin-labeled RNA sample, about $2.5 \times 10^8$ HeLa cells were collected and resuspended with 2.5 ml ice cold resuspension buffer (50 mM Tris-HCl pH 8.0, 150 mM NaCl). Cells were mixed with 2.5 ml 2x lysis buffer (50 mM Tris-HCl pH 8.0, 150 mM NaCl, 15 mM NaN₃, 1%(V/V) NP-40, 2 mM DTT, 2 mM PMSF, 2x protease inhibitor mix) and lysed for 5 min, and then centrifuged at 12,000 *g* for 20 min at 4 °C. Then 0.75 nmol biotinylated RNA with two 18 atom spacers (Dharmacon) were added to the supernatants and incubated for 2 hrs at 4 °C. Next, 50 μl Streptavidin-agarose beads (Thermo Fisher) were added into the mixture and incubated for 2 hrs at 4 °C with slow rotation. The beads were washed 3 times using 4 ml lysis buffer (50 mM Tris-HCl pH 8.0, 150 mM NaCl, 15 mM NaN₃, 0.5% NP-40, 1 mM DTT, 1 mM PMSF, 1x protease inhibitor mix), resuspended in 40 μl final volume, and mixed with 40 μl 2x SDS loading buffer. The proteins were then separated with a 4-20% ExpressPlus™PAGE Gel (GeneScript) and stained with coomassie blue. The gels were kept in 3% acetic acid for the further mass spectrometry analysis. The interested bands that contained candidate *trans*-factors were cut and analyzed by mass spectrometer.

**In vitro circRNA synthesis**. The circRNAs were generated through self-splicing of group I intron as described in the previous report[27]. Generally, linear RNAs were synthesized by in vitro transcription from linearized DNA templates using the RiboMAX™ Large Scale RNA Production System—T7 (Promega). DNA templates were digested through DNaseI treatment (at 37 °C for 15 min) after in vitro transcription. Then, RNAs were purified by the RNA Clean Up kit (Zymo). To circularize the RNA, it was heated to 70 °C for 5 min and then immediately placed on ice for 3 min, and then incubated in reaction buffer (2 mM GTP, 50 mM Tris-

HCl, 10 mM MgCl₂, 1 mM DTT, pH 7.5) at 55 °C for 15 min. After circularization, RNAs were purified by the RNA Clean Up kit (Zymo), and treated by RNase R (at 37 °C for 20 min). Finally, circRNAs were purified by SHIMADZU LC-20 series HPLC using a 7.8 × 300 mm size-exclusion column with particle size of 3.5 μm and pore size of 450 Å (Waters; part number: 186007643).

**Average hexamer frequency**. We analyzed the distributions of each type of hexamers using the average hexamer frequency, which was defined as the average occurrence for each hexamer from certain hexamer sets in various transcript regions. A sliding window was used to break each type of transcripts into overlapping hexamers (with 5-nt overlaps), and the frequency of each hexamer was calculated by its number of occurrences divided by the total number of all hexamers in the transcripts (mRNA or circRNA). The average hexamer frequency is the mean value of all hexamer frequencies in each type of hexamers (e.g., IRES-like hexamers, depleted hexamers, or all hexamers).

**Identification of circRNA-coded proteins**. All potential ORFs (i.e. three reading frames of the sense strand) encoded by the circRNAs were predicted. Firstly, each circRNA sequence was repeated four times to generate a concatemer sequence. Secondly, NTG start codon (non-ATG start is common in IRES-mediated translation) and TAA/TAG/TGA stop codon were determined in each concatemer sequence. Thirdly, each frame containing ≥20 aa was classified as a circORF. If there was no stop codon in the circORF, this ORF was predicted to generate an infinite protein product through rolling circle translation, defined as rolling circle ORF (rcORF).

Using an open search engine, pFind (v3.1.3, with default search that not allowed blind modification)[72], we searched two previous published human comprehensive proteome datasets (22,909,431 spectra)[42,43] against a combined database containing all UniProt human proteins (174,392) and the potential circRNA-coded peptides (177,907) across back-splice junctions from all three frames. The circRNAs were collected from circBase combining with the dataset of full-length circRNAs[20,33]. We selected positive mass spectra across back-splice junction using following thresholds: q < 0.01(spectrum level), peptides length ≥8 with a new sequence of at least 2 aa at either side of the back-splice junction, missed cleavage sites ≤3, allowing only common modifications (cysteine carbamidomethylation, oxidation of methionine, protein N-terminal acetylation, pyro-glutamate formation from glutamine, and phosphorylation of serine, threonine, and tyrosine residues).

A series of filters were used to remove the peptides that potentially from known proteins. (i) We combined a database containing all UniProt human proteins and potential circRNA-coded peptides from all three frames, and searched two comprehensive human proteome datasets against this combined database. The spectra mapped to known human proteins in UniProt were removed. (ii) For the remaining spectra that mapped to circRNA-coded peptides, we used Blastp 2.6.0+ to further remove the peptides that are homologues to known proteins (allowing two mismatches) in non-redundant human protein database. (iii) We finally used a strict cutoff to select positive spectra in which the circRNA-coded peptides were broken into fragment ions at both sides of back-splice junction.

For control peptides encoded by corresponding linear mRNAs, we selected the 5′ and 3′ splice junctions adjacent to the back-splicing junctions of all circRNAs, and used identical pipeline and cutoff to search the same MS-MS datasets for peptides encoded by the adjacent splice junctions. The resulting mass spectra supporting peptides across the canonical splice junctions adjacent to the circRNAs were further analyzed.

**Analysis of protein–protein interaction network**. To predict the protentional function of circRNA-coded proteins, we selected the host genes that include the translatable circRNAs (i.e. circRNAs contain the identified back-splice junction-coded peptides in mass-spectrometer data) for further analysis. These host genes were classified into two categories based on the circORF size: cORF with finite circORF, and rolling circle ORF (rcORF) with infinite circORF. To analyze the protein–protein interaction networks, we searched STRING 8 by both categories of the host genes with the parameters of high confidence, considering of all interaction evidences and discarding disconnected nodes. The resulting networks were clustered using MCODE tool (v1.6.1) in Cytoscape (v3.8.0). The function of each cluster was annotated using DAVID Gene ontology tool.

**Position distribution of IRES-like hexamers in different regions of mRNAs**. To analyze the distribution of short IRES-like elements in mRNAs (near start codon and stop codon), we selected the adjacent regions (±300 nucleotides) of the annotated start codon or stop codon from all the protein-coding transcripts (RefGene of hg19). Then, a sliding window of 6-nt with step size of one nucleotide was employed from the 5′ to 3′ end on all transcripts. In each window, the hexamer frequency was calculated as the total occurrences of the hexamers sets (enriched hexamers or depleted hexamers) in all the sequences divided by the total number of all hexamers in these sequences.

**Correlation analysis of 18S rRNA regions and identified hexamers**. Short sequences were reported to function as IRESs by pairing with certain regions of 18S rRNA (i.e. "active region"). To test whether the IRES-like hexamers are similar as

the 18S rRNA active region, we examined the correlation between 18S rRNA regions (active region and inactive region) and identified hexamers (enriched hexamers and depleted hexamers) in this study. Heptamers ($p$ value < 0.05) of 18S rRNA active region and inactive region were extracted from published datasets[19], and each heptamer was split into two hexamers. The distribution of these hexamers was calculated by their enrichment score in Supplementary Data 2.

**Statistics and reproducibility**. All experiments were repeated three times independently. The samples from the same batch derive from the same experiment and that gels/blots were processed in parallel. Error bars represent mean ± SD. P-values were calculated with two-sided Student's $t$ test.

**Reporting summary**. Further information on research design is available in the Nature Research Reporting Summary linked to this article.

## Data availability
Four published circRNA datasets (Figs. 2a, 4a) were retrieved from circBase (http://circbase.org/). The full-length circRNA dataset (Fig. 4b, d) was obtained from the published ribominus RNA-seq data that was generated from the RNase R treated RNAs of HeLa cells (BioProject database of Genbank, accession number PRJNA266072). Two human comprehensive proteome datasets (Fig. 4d–g, and Supplementary Fig. 5a, b) were obtained from Bekker-Jensen, DB, et al, and Kim, MS, et al. & Pinto, SM, et al., which in turn rely on freely available data obtained from PRIDE Archive (accession number: PXD004452, and PXD000561) The RNA-seq data generated in this study have been deposited in the GEO database under accession code GSE152560. Source data are provided with this paper.

## Code availability
The custom scripts used in this study are available in Github repository (https://github.com/rnasys/IRES-like-elements).

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

## Acknowledgements

The authors want to thank Dr. Hao Chi for his support in identifying circRNA-coded peptides using open pFind(v3.1.3), Dr. Fangqing Zhao for sharing the unpublished full length circRNA dataset, Ms. Yue Hu for help in analyzing RBP binding data, and Drs. Reinhard Lührmann and Xiaoling Li for useful discussions and comments. We thank the National Center for Protein Science for LC-MS/MS analysis in the identification of the *trans*-factors. This work is supported by the National Key Research and Development Program of China (MOST grant 2021YFA1300503), and the National Natural Science Foundation of China to Z.W. (91940303, 32030064, and 31730110), Y.Y. (91753135, 31870814), and X.F. (32100430). Z.W. is also supported by the type A CAS Pioneer 100-Talent program. Y.Y. is also sponsored by the Youth Innovation Promotion Association CAS and Shanghai Science and Technology Committee Rising-Star Program (19QA1410500). X.F. is sponsored by the fellowship of China Postdoctoral Science Foundation (2020M681437).

## Author contributions

Conceptualization, Z.W. and Y.Y; Methodology, X.F., Y.Y., and Z.W.; Software, X.F.; Experiments, X.F., Y.Y. C.C.; Writing, X.F., Y.Y., and Z.W.; Funding Acquisition, X.F., Y.Y., and Z.W.

## Competing interests

Z.W. and Y.Y. has co-founded a company, CirCode Biotech, to commercialize the application of circular RNA as template of protein production/expression, and applied a patent to use circRNA as a gene expression platform. The other authors declare no competing interests.
