## [Peer Review File · Nature Communications]

Title: Pervasive translation of circular RNAs driven by short IRES-like elementsREVIEWER COMMENTS

Reviewer #1 (Remarks to the Author):

The study by Fan X. and Yang Y. et al identified 97 IRES-like hexamers that can drive cap-independent circRNA translation through a GFP-based screen of random decamers. These IRES-like hexamers were found to be enriched in circRNAs and the ability to promote circRNA translation was validated experimentally through several approaches. Some trans-acting protein factors may recognize these IRES-like motifs and play a role in promoting translation. By searching in publicly available mass spectrometry databases, the authors identified hundreds of peptides that may be encoded by circRNA although the overall abundance of peptides is low. 50% circRNA with identified mass spectrometry peptide may undergo rolling circle translation, highlighting the unique properties of circRNA translation. The study revealed novel IRES-like motifs which enforces the notion that circRNA may be pervasively translated. The identified trans-acting factors may further contribute to understanding the mechanism of cap-independent translation. While the study is overall insightful, more rigorous controls should be included in some experiments to solidify the conclusions.

Major comments:

The plasmid-based overexpression system used by the authors has been reported to produce false-positive signals arising from the rolling circle transcription of the plasmid, and conclusions based on expression constructs need to be drawn with caution:

<https://www.life-science-alliance.org/content/2/3/e201900398>

<https://doi.org/10.1093/nar/gkaa704>

<https://doi.org/10.1016/j.jymeth.2021.02.007>

During splicing, the downstream splice donor couples to the upstream splice acceptor to form a conventional linear splicing product that is indistinguishable from backsplicing, which will give rise to the exact same protein product if translated. This phenomenon is more common in HEK293T cells used in this study. However, linearizing the plasmid before transfection or using HEK293 cells that do not express the large T antigen will significantly reduce this occurrence.

Although the authors used 293-Flp-In stable cell for verification, in Fig S2C, the signal of the “cGFP with m6A” sample is almost invisible, while other samples still maintain similar abundance. This happens to be an indication that the most of the signal observed by the author might be peculiar to the plasmid-based overexpression.

Please include FACS gating on wild type 293T cells without reporter transfection to justify the gating of GFP negative cells. Auto-fluorescence of cells might be a concern.

What were the positive and negative controls in the screen?

In Figure S1G, circRNA with depleted motifs especially GUGUCA seemed to increase luciferase signals as well and the levels appeared to be similar to circRNA with AUAUUAU. Please include some statistics to highlight that depleted motifs do not increase luciferase activity while IRES-like motifs do.

In Figure 2B, it is inexplicable that rcGFP in the sixth lane can also produce rolling circle translation products. The author’s explanation is "through an internal sequence function as an IRES", which requires proper experimentation to confirm, otherwise, the circGFP reporting system may be unreliable, as well as the downstream conclusions.

In Fig3D, why did the authors choose ELAVL1, PABPC1, HNRNPA1 and HNRNPU for the experiment? None of them were present in every sample in Fig3B. Instead, some proteins shared by these samples (such as RBMXL1, RBMX) were ignored.

circRNA-encoded peptides were searched in a mass spec database containing 30 tissues and 6 cell lines. The circRNA annotation used was also a compilation of data from multiple cell lines. circRNAs are known to have tissue-specific expressions. For the circRNAs with identified peptides, are they expressed in the source tissue/cell line where the peptide originated? Line 311-312 mentioned that 60-80% of peptides were identified in a single sample. Are the corresponding circRNAs expressed in the sample? False positive results and peptide low abundance are a concern here.

In Figure S5E, cGFP fusion proteins with circRNA-coded C-terminal tails were compared to wild type cGFP. However, fusion of additional peptides, even epitope tags, can sometimes affect protein stability. Additional negative controls such as fusion of N-terminal peptides or shuffled peptides should be included before drawing conclusions.

Minor comments:

Please bold subheadings G and H in the legend of Figure 1.

The images of the GFP and GAPDH RNA blots of Fig 3D are from the same membrane with different degrees of exposure.

Line 459, "deference" should be changed to "difference".

Reviewer #2 (Remarks to the Author):

It's a potentially interesting study, but I'm unsure how relevant their findings are from a biological context. It's difficult for me to assess their MS data, but they don't seem super compelling to me. However, reading their conflicts of interests statement, it looks like they may have started a company to commercialize their findings to overexpress proteins in mammalian cells. When their data are taken in that context, the flow of the paper is much more logical. It feels like they've taken a series of experiments to develop a technology and tried to frame them in a biological context, rather than the more appropriate biotech framing.

I also have some reservations about their experiments and conclusions.

In Figure 1, the authors used a screen to identify sequence elements that promote translation of circRNAs. They created a DNA library where programmed backsplicing of the transcribed mRNA leads to circularized RNA (circRNA) with the sequence TAA-NNNNNNNNNN-ATCATGG (essentially, the stop codon, the 10 random nts, a 3 nt spacer, and then the start codon that precedes GFP). Following transfection of the DNA library into human cells, they used FACS to sort pools of cells that corresponded to different levels of GFP expression (none, low, mid, high). High-throughput sequencing and enrichment analyses were then used to identify nucleotide sequences that yield GFP fluorescence, presumably via translation of the circRNA.

One primary concern is that many of the enriched sequences (i.e. those that promote GFP fluorescence) resemble a cleavage and polyadenylation signal (canonically, AAUAAA, with single-nucleotide deviations also allowed; see, <https://www.nature.com/articles/s41576-019-0145-z.pdf?origin=ppub>). While their library construction was clever, it is possible that some (or, more worrisome, many) of the sequences they have identified as IRES-like elements may instead lead to cleavage of the newly-formed circRNA (via the programmed backsplicing) into a now linear mRNA with a start codon near the 5' terminus and the re-formed GFP coding sequence downstream, now potentially ending with a poly(A) tail. Thus, their GFP signal instead could arise from such unintended, linearized mRNAs that occur at low efficiency. The authors should alleviate this concern experimentally, such as by sequencing the ends of the linear mRNAs detected in Fig S1E with enriched and depleted hexamers.

A second concern is the relevance of the IP-MS/MS experiments to identify proteins that bind the “IRES-like” hexamers. The authors used a linear RNA 20 nt in length that encodes tandem repeats of hexamers of interest. I understand why this approach was taken initially, but it removes the sequences from the circRNA context, which the authors data suggest is required for their “IRES-like” activities (Supp Fig 7). Indeed, the sequence bound by PABPC1 (AAAAGx3) would resemble a short poly(A) tail when present in the linear bait RNA, which is the canonical substrate of the protein. It would strengthen the manuscript if the authors were able to demonstrate that the identified proteins bind circRNAs that contain the particular enriched hexamers, and don't bind circRNAs that lack the sequence.

Relatedly, as potential functional validation of the identified proteins, the authors used a modified tethered-function assay where an engineered PUF protein domain was used to target proteins of interest to designed circRNAs that encode GFP. Here they observed the specific increase in GFP translation when PABPC1, and more modestly HNRNPU, was tethered. This finding recapitulated the effect of the “neo-activator” on cyclin-B1 mRNA reported in Figure 6 of Campbell et al (<https://www.nature.com/articles/nsmb.2847>), but now in the context of a circRNA, which potentially supports their model. However, the more compelling experimental validation where the authors tested whether overexpression of PABPC1 could enhance circRNA translation – without tethering – is missing important controls and/or experimental details. Unless I've missed it, the authors only examined the effect of PABPC1 overexpression on a circRNA with the A10 sequence. To ensure specificity of the enhancement effect, the authors should examine whether PABPC1 overexpression enhances translation of a circRNA with a depleted sequence. Similarly, it is unclear how loading of the western blots was normalized, making it difficult to assess whether PABPC1 overexpression also increased translation of the GAPDH loading control.

Minor points:

1. “Interestingly, the deletion of “AUC” sequence reduced translation in 4 out of 6 enriched hexamers tested but increased translation driven by one hexamer (poly-A sequences).” Is this an effect of Kozak context on translation efficiency? The “AUC” would provide optimal -3 context (the A). However, related to the point above, that same A would complete the canonical AAUAAA cleavage & polyadenylation

signal, particularly for the “UAAUAA” hexamer sequence, where the largest decrease in translation is observed upon deletion of the spacing “AUC”. To discriminate these potential effects, the authors could substitute the A in the spacing AUC for G, which would provide near optimal Kozak context.

Reviewer #3 (Remarks to the Author):

Fan et al., presents an interesting work on circRNAs. The authors argue that circRNAs-coded peptides are actually translated, and they are cell type, tissue, or biological process specific. To prove their presence, the authors adopted mass spectrometry based peptide identification, on which my review will focus.

The datasets used for the search are two published human datasets. And the database was constructed from the three frames of the published circRNAs sequences. Peptides were searched using pFind with FDR threshold (or q-value threshold) of 1%. Further stringent filtration to prevent false peptide matches from non-circRNA region was also used. After the search, more than 2,000 peptides were identified and analyzed.

Overall, the peptide ID procedure is straight and considerably stringent. I do not see much possibility that those 2,000 peptides represent false positives or those from non-circRNA area. However, the authors could address the following points for better understanding of MS-based readers.

- 1) It would be nice if the authors specify a) the size of datasets (#spectra), b) the size of constructed circRNA DB along with the appended human DB (in terms of #proteins or length of amino acid sequences). They could be on Methods section.
- 2) In supplement table 4, it would be nice to add the explanation on the remaining columns. Some are explained but others are not. What is the last column meaning (separate FDR)?
- 3) Maybe I missed, but it is not well described that the ID from human protein DB are filtered out (only circRNA hits were retained).
- 4) Out of 2,000 ID spectra, how many were identified in the previous publication? If this number is low, proteomics readers would be very intrigued because such non standard peptides are always of a great interest for them (often called dark matter..).
- 5) pFind enables blind modification search as far as I know. Is this search also allowed blind modification?
- 6) The annotations shown in Fig 4 a) look almost perfect, and delivered the point of the paper very well. But it would be nice if some more examples could be shown in the supplement figures.
- 7) is q-value used in search: is it peptide level or spectrum level?

Overall, this paper looks very interesting and the analysis was done very well, in particular in terms of MS-based peptide analysis.

REVIEWER COMMENTS

Reviewer #1 (Remarks to the Author):

The study by Fan X. and Yang Y. et al identified 97 IRES-like hexamers that can drive cap-independent circRNA translation through a GFP-based screen of random decamers. These IRES-like hexamers were found to be enriched in circRNAs and the ability to promote circRNA translation was validated experimentally through several approaches. Some trans-acting protein factors may recognize these IRES-like motifs and play a role in promoting translation. By searching in publicly available mass spectrometry databases, the authors identified hundreds of peptides that may be encoded by circRNA although the overall abundance of peptides is low. 50% circRNA with identified mass spectrometry peptide may undergo rolling circle translation, highlighting the unique properties of circRNA translation. The study revealed novel IRES-like motifs which enforces the notion that circRNA may be pervasively translated. The identified trans-acting factors may further contribute to understanding the mechanism of cap-independent translation. While the study is overall insightful, more rigorous controls should be included in some experiments to solidify the conclusions.

We thank the reviewer for appreciating novelty of this work, and have addressed the concerns of the control experiments with additional analyses.

We hope to first address a minor point of this reviewer “2) The images of the GFP and GAPDH RNA blots of Fig 3D are from the same membrane with different degrees of exposure”, because we think this is the most important concern.

First of all, I think it is fair to question the data integrity on any results when they look “too good”, **and we really appreciate that this reviewer directly pointed it out rather than second guessing.** However as we will show below, the data has not been manipulated.

The reviewer is correct in that the two blots look similar to each other, but they are actually different bands rather than the same blots with different exposure, which can be clearly distinguished from each other as judged by the little spots in the backgrounds (marked in red circles, see below). In this experiment, we mixed the samples of two different RT-PCR reactions (with GAPDH and GFP specific primers) together and run them on the same gel to save time and to reduce potential inconsistency from repeat exposure of different gels. Because the GAPDH and GFP bands run on different positions, we can clearly distinguish them while reducing technical variations between parallel samples. As a result, the shapes of bands look similar because they are from the same gel. To clarify this, we have now included the original blots in below for a comparison (see below). We also included additional text in the supplementary methods (page 5) to clarify how we analyzed the RT-PCR samples.

Fig. 3D bottom panel. Comparison of the original gel in its entirety (top) and fraction showing the RNA levels of GFP and GAPDH (bottom). The mock controls were the same in the three sets of experiments (cells transfected with circRNA reporter only) and thus the first one was included for the final figure.

To high-light differences between the GFP bands and the GAPDH bands, we also arbitrarily adjusted the brightness and contrast, and thus the small differences in the background can be visualized more clearly.

Major comments:

- 1) The plasmid-based overexpression system used by the authors has been reported to produce false-positive signals arising from the rolling circle transcription of the plasmid, and conclusions based on expression constructs need to be drawn with caution: <https://www.life-science-alliance.org/content/2/3/e201900398>
<https://doi.org/10.1093/nar/gkaa704>
<https://doi.org/10.1016/j.ymeth.2021.02.007>

During splicing, the downstream splice donor couples to the upstream splice acceptor to form a conventional linear splicing product that is indistinguishable from back-splicing, which will give rise to the exact same protein product if translated. This phenomenon is more common in HEK293T cells used in this study. However,

linearizing the plasmid before transfection or using HEK293 cells that do not express the large T antigen will significantly reduce this occurrence.

We are aware of these reports regarding the potential artifact from linear mRNA, which has been discussed and supported with indirect evidences (i.e., lack of translation or weak translation in circRNA reporters and small amount of translation in mutated reporters). Compared to the two circRNA reporters used in these papers, our GFP-based back-splicing and translation system is different in that strong intronic complementary sequences were used to increase back-splicing efficiency (compared to other unintended splicing events), and thus we observed robust translation with the necessary controls.

In addition, we conducted detailed analysis on the potential contributions from linear mRNA in the circRNA translation reporters, and are in the process to prepare another manuscript addressing these issues. Some of the data were include below to address the reviewer's concern of false-positive translation from linear mRNAs (rebuttal Fig. R1), and additional information and discussion will be included in the separate manuscript.

Fig. R1. Examine the potential contribution of linear RNA artifacts to circRNA translation. (A) Schematics of the contaminated linear RNAs that might be generated from the rolling circle transcription (i.e. Poly-A leakage) or trans-splicing. **(B)** Step-wise addition of poly-A sites do not decrease circRNA translation. **(C)** Linearization of expression plasmid before transfection do not affect circRNA translation. **(D)** Determine the level of trans-spliced linear RNAs vs. back-spliced circRNA using RNA-seq of cells co-transfected with two versions of reporter plasmids. **(E)** Determine the GFP protein levels using cells transfected by different versions of plasmids that produce proteins with different N- and C- terminal tags.

As shown in the model above (rebuttal Fig. R1A), the contaminated linear mRNAs can be generated from two pathways: (1) The rolling circle transcription (i.e., leakage of polyA signal) may generate a very long RNA concatemer, which could then be spliced into a long linear mRNA. (2) The pre-mRNAs may undergo *trans*-splicing (very

rare in human cells) to produce a long linear mRNA. In both cases, the resulting linear mRNA will have the exactly same sequence as the back-splice junction and may be translated with the internal ATG to initiate translation to produce functional GFP (rebuttal Fig. R1A).

To examine the contribution from transcription readthrough, we first generated a series of plasmid reporters with 1-3 polyadenylation sites at the downstream of the split-GFP gene, where the additional poly-A sites will reduce readthrough transcription (rebuttal Fig. R1B). However, we found that the additional number of poly-A sites did not reduce the GFP production, suggesting that the transcription read-through is not a major contributor to GFP production in this system.

In addition, as suggested by the reviewer, we have linearized the plasmids with two different restriction enzymes (MfeI and SphI), and found no significant difference in the GFP translation (rebuttal Fig. R1C), suggesting that the majority of GFP translation is from the circRNAs. This data is consistent with what we reported earlier using the same linearized plasmid¹.

To examine the contribution of *trans*-splicing in this reporter system, we generated two mutated reporters by introducing deletions in the 5' or 3' half of the GFP exon, which will cause frame-shift for the back-splicing products but maintain the reading frame for the *trans*-splicing products (rebuttal Fig. R1D, left). The first reporter (FP-del/G) has a -2 deletion in the C-terminus of the GFP ORF, and the second reporter (FP/G-del) has -1 deletion in N terminus of GFP. Therefore the back-splicing product will have a -2 or -1 frame-shift for GFP. However, when we co-transfect these two reporters, the *trans*-splicing products from the two pre-mRNAs will restore reading frame of GFP. Using this design, we co-transfected two mutated reporters into 293T cells and examined the RNA products with RNA-seq and the protein products with western blot. We found that ~95% of junctions were generated from back-splicing rather than *trans*-splicing (rebuttal Fig. R1D, right), and there was no detectable GFP proteins. This experiment suggests that the *trans*-splicing of pre-mRNAs is very rare in the over-expression of our reporter system, and the observed proteins are predominately produced from circRNA translation rather than *trans*-splicing products.

Furthermore, we used an independent experiment to access the potential effect of *trans*-splicing products. We labeled the N- and C-terminal of split-GFP reporter with HA and Flag tag respectively, and generated three different reporters with the combination of the epitope tags (rebuttal Fig. R1E). We co-transfected the two reporters with either N-terminal or C-terminal tag into 293T cells, and pull-down the proteins with HA-bead or Flag-bead for western blot detection. We found that the proteins produced from this experiment has either N-terminal or C-terminal epitope tag but not both (rebuttal Fig. R1E, last panel on the bottom), suggesting that these proteins are produced from back-spliced circRNAs rather than the *trans*-spliced linear mRNAs. Collectively, these experiments confirmed the reliability of our system and largely ruled out the contribution from linear RNA artifact.

Collectively, our data suggested that most of the proteins generated from this circRNA reporter system are indeed translated from circRNAs rather than the linear RNA contaminates. Similar experiments with split-GFP based circRNA translation

reporters were also independently confirmed by studies from other groups including two recent studies from the labs of Howard Y Chang and Ling-Ling Chen ^{2,3}.

Finally, as suggested by the reviewer, we also repeated the key experiments of circRNA translation in two additional cell lines that do not express large T antigen, and obtained the similar results. These results were now included in the Fig. S2C of revised manuscript and page 9 of main text.

Fig. S2C. Testing translation of circRNA reporters in additional cell types (HEK293 cells and HCT116 cells). Similar to experiments in Fig. 2B, the circRNA plasmids were transfected into indicated cells, and samples were analyzed by western blot at 2 days after transfection (right panel). The green arrow indicates the full-length GFP, and the smears at the top of gels indicate rolling circle translation products. The bar graph represents the quantification of GFP protein relative to GAPDH. The protein level was also normalized to the RNA level (n = 3, mean ± SD).

2) Although the authors used 293-Flp-In stable cell for verification, in Fig S2C, the signal of the “cGFP with m6A” sample is almost invisible, while other samples still maintain similar abundance. This happens to be an indication that the most of the signal observed by the author might be peculiar to the plasmid-based overexpression.

We agree that all the bands in the 293-Flp-In stable cells are weaker than the transience transfected samples (not just the “cGFP with m6A” was weaker than transient transfection), which is due to the fact that only a single copy of reporter was inserted in the genome using Flp-In system. The m6A has been previously reported by us and other groups to drive the translation of circRNAs, and thus we used this as a positive control. The main point of this figure is to confirm the rolling circle translation in the two rcGFP constructs, which can be clearly visualized in both Fig. 2B and Fig. S2D (updated from old Fig. S2C) and were pretty strong compared to the positive control of m6A. Because the rcGFP plasmids were also stably inserted into cells, this is unlikely a “peculiar” observation from plasmid-based over-expression. Please also see our control experiments from our reply to the point #1.

3) Please include FACS gating on wild type 293T cells without reporter transfection to justify the gating of GFP negative cells. Auto-fluorescence of cells might be a concern. What were the positive and negative controls in the screen?

As suggested, we have now included the FACS gating on wild type 293T cells (negative control) and the cells transfected with GFP expression reporter as the positive control to set the threshold of cell sorting (updated Fig. S1B and Fig. 1B). Because this is an unbiased screen of random sequence library, we applied a pretty stringent gating to set up the screen (i.e., only used the cells with medium and high

GFP signals). In addition, we also tested the circRNA reporters inserted with short poly-A and poly-G sequences (Fig. S1A), and used them as the positive and negative controls in the screen. Such information is now included in the revised methods (page 31 of methods and page 2 of supplementary methods).

Fig S1B

Fig. S1B. Stepwise gating of single live cells in FACS. The green fluorescence of the cells after such gating is shown. The wild-type 293T cells and the cells transfected with the pEGFP-C1 (an EGFP expression vector) were used as negative and positive controls (GFP-ctl, GFP+ ctl) for the gating of green fluorescence during cell sorting (Fig. S1B).

Fig.1B. Flow-cytometry analysis of cells transfected with circRNA reporter containing the random 10-mer library. The transfected cells were classified into four groups based on their GFP fluorescence (GFP negative cells and cells with low, medium or high GFP signals). The cells with medium and high fluorescence were sorted as “green cells”.

4) In Figure S1G, circRNA with depleted motifs especially GUGUCA seemed to increase luciferase signals as well and the levels appeared to be similar to circRNA with AUAUUA. Please include some statistics to highlight that depleted motifs do not increase luciferase activity while IRES-like motifs do.

As suggested, we have included the statistics information in the revised Fig. S1G and the figure legend. The result shows that newly identified short IRES-like elements increase luciferase signals significantly compared to depleted elements in both 293T cells ($p=0.03$, Welch's t-test) and HCT116 cells ($p=0.004$, Welch's t-test).

Fig. S1G. Translation of in vitro synthesized circRNAs. CircRNAs of luciferase containing known IRESs and newly identified IRES-like or depleted hexamers were generated through self-splicing of group I intron. These circRNAs were transfected into 293T and HCT116 cells. 24 hours after transfection, the cells were lysed for luminescence measurement using microplate reader (mean \pm SD, $n = 3$ independent experiments). The P values of the differences between newly identified short IRESs and depleted motifs was calculated by Welch's t-test.

5) In Figure 2B, it is inexplicable that rcGFP in the sixth lane can also produce rolling circle translation products. The author's explanation is "through an internal sequence function as an IRES", which requires proper experimentation to confirm, otherwise, the circGFP reporting system may be unreliable, as well as the downstream conclusions.

The reviewer may be confused by the main point here, probably because we did not sufficiently explain the rationale and experimental design.

The result of unbiased screening shown that there are 97 AU-rich hexamers that may function like an IRES to initiate cap-independent translation. For all hexamers, there are four possible bases in each of the six positions, and thus there would be 4^6 (=4096) all hexamers in the entire sequence space. Theoretically, about 2% of all hexamers (97/4096 \approx 2%) are IRES-like elements. In another word, there would be one IRES-like element in every 50 hexamers (97/4096). Since >99% circRNAs are longer than 100-nt (Fig. S2A), most circRNAs should contain internal IRES-like short elements **by chance**, even in the putative ORF region. This conclusion is surprising because most circRNAs may not require an additional known IRES to drive its translation.

To test this surprising hypothesis, we used the rcGFP (717nt), which is a circRNA only contains a GFP ORF without stop codon, to examine the translation of rcGFP through an internal hexamer. The GFP gene contains 4 IRES-like hexamers in its coding region, and thus should be able to translate from these internal IRES-like elements. As expected, our results showed that this ORF-only circRNA lacking a stop codon can indeed be translated into large protein through rolling circle translation. To further test this hypothesis, we also generated two additional circRNAs containing luciferases ORF (Rluc (933nt) and Fluc (1650nt)) but lacking a known IRES, and obtained the similar results of circRNA translation without requirement of known IRES (Fig. 2C and Fig. S2E-G). We have now included additional text in revised manuscript to provide more detailed explanation (page 9 of main text).

6) In Fig3D, why did the authors choose ELAVL1, PABPC1, HNRNPA1 and HNRNPU for the experiment? None of them were present in every sample in Fig3B. Instead, some proteins shared by these samples (such as RBMXL1, RBMX) were ignored.

The purpose of this experiment is to identify *trans*-acting factors that specifically bind to IRES-like elements, therefore we chose the well-known factors: ELAVL1, PABPC1, HNRNPA1, and HNRNPU that are known to specifically bind AU rich elements. Because the RBMXL1 and RBMX have been identified in multiple samples regardless of bait RNAs, we suspect that it could be an artifact of RNA-IP because they can bind any RNA with low specificity (and thus did not focus on these two proteins). We have now included additional text to clarify the logic for how we choose these proteins for further test (page 12 of main text).

7) circRNA-encoded peptides were searched in a mass spec database containing 30 tissues and 6 cell lines. The circRNA annotation used was also a compilation of data

from multiple cell lines. circRNAs are known to have tissue-specific expressions. For the circRNAs with identified peptides, are they expressed in the source tissue/cell line where the peptide originated? Line 311-312 mentioned that 60-80% of peptides were identified in a single sample. Are the corresponding circRNAs expressed in the sample? False positive results and peptide low abundance are a concern here.

We agree with the reviewer that, in an ideal scenario, using matched sets of RNA-seq and proteomics data would improve the search specificity and reduce the FDR. However, the individual proteomic datasets all have low coverage and variable data quality, and thus we could not find a matched sets with good coverage of both types of data for a reliable identification. In addition, a comprehensive dataset of circRNAs profiling in different human tissues is lacking, and thus we have to merge all circRNAs from different cell lines/tissue to identify the circRNA-coded protein productions.

Since the main goal of this study is to identify all possible products from translatable circRNAs, we first used two human proteome datasets with deep coverage from multiple tissues and cell lines. We searched these comprehensive datasets against a combined database containing all UniProt human proteins and all potential circRNA-coded peptides, which allow us to identify the circRNA encoded proteins with high sensitivity. From the initial search results, we applied a series of filters to improve the specificity.

It is not surprising that 60-80% of peptides were identified in a single sample, because recent studies showed limited overlaps of circRNAs from different cell lines/tissues and different databases, or identified by different tools. These data suggest that most circRNAs are expressed in cell/tissue specific fashion. In addition, we have applied a series of stringent filters and statistic test in the analysis of proteomic data (please see the comments and response to the third reviewer), which will reduce the false positive rate. We included additional text to discuss the concerns of false positive identifications (page 15), and think additional analysis in the future with more comprehensive datasets will further improve the sensitivity and specificity.

8) In Figure S5E, cGFP fusion proteins with circRNA-coded C-terminal tails were compared to wild type cGFP. However, fusion of additional peptides, even epitope tags, can sometimes affect protein stability. Additional negative controls such as fusion of N-terminal peptides or shuffled peptides should be included before drawing conclusions. We fused the C-terminal tails because these are the circRNA-coded “new” sequences distinct from the canonical proteins. As suggested, we have now included additional negative controls by fusing the N-terminal peptide of the same circRNA-coded protein or a V5-epitope tag to cGFP, and found no obvious degradation of these control cGFP fusion proteins as judged by the protease inhibitors, supporting our main conclusion. We have now modified the Fig. S5E and revised the text in page 20. “Moreover, the expression level of the fusion proteins with circRNA-coded tails increased upon the inhibition of proteasome degradation by MG132 treatment, whereas the GFP without such tails or with control peptide tails

(N-terminal peptides of the same gene or a V5 epitope tag) were essentially unaffected (Fig. S5E)."

Fig. S5E. CircRNA-coding tails induce rapid protein degradation through proteasome pathway. The circGFPs with different circRNA-coded tails (circCRKL-tail, circMAN1A2-tail, or cRBM4-tail, top panel) and N-terminal peptide of the same circRNA-coded protein (CRKL N-terminal or MAN1A2 N-terminal, bottom panel) or V5-epitope tag were transfected into 293T cells. Then the transfected cells were treated with 10 μ M MG132 for 2 hours, or 10 μ M chloroquine for 4 hours before cell collection.

Minor comments:

1) Please bold subheadings G and H in the legend of Figure 1.

We have revised this with correct font.

2) The images of the GFP and GAPDH RNA blots of Fig 3D are from the same membrane with different degrees of exposure.

This figure is not the membrane with different degrees of exposure, and we have the original gel to prove it. Please see our response in the beginning (before response to all other points).

3) Line 459, "deference" should be changed to "difference".

We have fixed these mistakes and will do a careful proofread on the entire text.

Reviewer #2 (Remarks to the Author):

It's a potentially interesting study, but I'm unsure how relevant their findings are from a biological context. It's difficult for me to assess their MS data, but they don't seem super compelling to me. However, reading their conflicts of interests statement, it looks like they may have started a company to commercialize their findings to overexpress proteins in mammalian cells. When their data are taken in that context, the flow of the paper is much more logical. It feels like they've taken a series of experiments to develop a technology and tried to frame them in a biological context, rather than the more appropriate biotech framing.

We appreciate the insightful comments from this reviewer. Nowadays it is not uncommon for the investigators to seek commercial applications based on earlier findings from basic research. In our case, this study was designed toward the understanding of the translation initiation of circRNAs in cells. The results of this study have important biological impact, as they provided mechanistic insight on how circRNA can be translated and expanded the scope of circRNA translation. Only afterwards we found that the related knowledge may have application value and started the company.

I also have some reservations about their experiments and conclusions.

1) In Figure 1, the authors used a screen to identify sequence elements that promote translation of circRNAs. They created a DNA library where programmed backsplicing of the transcribed mRNA leads to circularized RNA (circRNA) with the sequence TAA-NNNNNNNNNN-ATCATGG (essentially, the stop codon, the 10 random nts, a 3 nt spacer, and then the start codon that precedes GFP). Following transfection of the DNA library into human cells, they used FACS to sort pools of cells that corresponded to different levels of GFP expression (none, low, mid, high). High-throughput sequencing and enrichment analyses were then used to identify nucleotide sequences that yield GFP fluorescence, presumably via translation of the circRNA.

One primary concern is that many of the enriched sequences (i.e. those that promote GFP fluorescence) resemble a cleavage and polyadenylation signal (canonically, AAUAAA, with single-nucleotide deviations also allowed; see, <https://www.nature.com/articles/s41576-019-0145-z.pdf?origin=ppub>). While their library construction was clever, it is possible that some (or, more worrisome, many) of the sequences they have identified as IRES-like elements may instead lead to cleavage of the newly-formed circRNA (via the programmed backsplicing) into a now linear mRNA with a start codon near the 5'terminus and the re-formed GFP coding sequence downstream, now potentially ending with a poly(A) tail. Thus, their GFP signal instead could arise from such unintended, linearized mRNAs that occur at low efficiency. The authors should alleviate this concern experimentally, such as by sequencing the ends of the linear mRNAs detected in Fig S1E with enriched and depleted hexamers.

The reviewer raised an interesting possibility, however may be confused about the

screen procedure. In order to recover the IRES-like elements enriched in “green cells”, we have to purify the total RNA from the green cells and then use RT-PCR to amplify **across** the insertion region with primer pairs on GFP. Therefore, if the inserted sequences cause the cleavage and poly-adenylation on circRNA, we would not be able to recover these sequences because the RNA would be broken. As a result, the sequences that cause circRNA cleavage would be **depleted** (rather than enriched) in the following-up steps of RNA-seq library recovery. Consistently, we have not detected any cleavage product or re-ligated RNAs in Northern blot. We have now modified the diagram in Fig. 1A to clarify this (page 5 and legend of Fig. 1A).

Fig. 1A. Schematic diagram for screening short IRES-like elements. Random decamers were inserted into pcircGFP-BsmBI reporter, and the resulting library was transfected into 293T cells and sorted by FACS. The green cells were collected and the inserted sequences were sequenced using high-throughput sequencing. The RT and PCR primers for RNA-seq library production were indicated by blue arrows. The hexamers enriched in green cells were identified by computational analysis (also see method).

In addition, the canonical poly-A signal required additional GC-rich motif at the downstream of “AAUAAA” consensus sequence, and the cleavage site is located at ~20 nt at the downstream of AAUAAA. Therefore, even if the “AAUAAA” caused the circRNA cleavage (unlikely due to lack of downstream GC-rich motif), the cleavage site would be inside the GFP ORF after the initiation ATG codon, which would destroy the ORF (i.e., no functional GFP will be produced, see rebuttal Fig. R2 below).

Fig. R2. Schematic of the circRNA cleavage and recapping/re-polyA by “AAUAAA” consensus signals (indicated as white box). The start and stop codons were marked with dark green and red lines, respectively.

Nevertheless, we still conducted 5'-RACE experiments in order to map the 5' end of the RNA products generated from the over-expression plasmids. Using GFP specific primers and SMARTer RACE 5'/3' Kit, we could not obtain clear and reliable band from RACE experiments, except for an artifact band corresponding to the pre-mRNA with part of the intron being spliced out (Fig. R3). The fact that we could not detect the cleavage and recapping products suggested that this is indeed a very rare

event. We have now included additional text in the discussion to address this concern (page 25-26).

Fig. R3. Schematic of the 5'-RACE to identify the end of circRNA cleavage product. We did not identify any band at the expected position, and the only band observed is corresponding to the pre-mRNA with part of intron being spliced out.

2) A second concern is the relevance of the IP-MS/MS experiments to identify proteins that bind the “IRES-like” hexamers. The authors used a linear RNA 20 nt in length that encodes tandem repeats of hexamers of interest. I understand why this approach was taken initially, but it removes the sequences from the circRNA context, which the authors data suggest is required for their “IRES-like” activities (Supp Fig 7). Indeed, the sequence bound by PABPC1 (AAAAAGx3) would resemble a short poly(A) tail when present in the linear bait RNA, which is the canonical substrate of the protein. It would strengthen the manuscript if the authors were able to demonstrate that the identified proteins bind circRNAs that contain the particular enriched hexamers, and don't bind circRNAs that lack the sequence.

As suggested, we performed RNA-IP using PABPC1 as example, and examined the binding between RBPs to circRNAs containing the enriched or depleted hexamers. As expected, we found that the identified RBP indeed binds more tightly to the circRNAs containing IRES-like short sequences compared to the depleted hexamers. This new data has now been included as Fig. S3C and described in page 13 of revised manuscript.

Fig. S3C. The association of PABPC1 with circRNAs containing IRES-like hexamers and depleted motifs was tested by RIP analysis using Anti-FLAG M2 antibody. After IP, the presence of PABPC1 in the beads was confirmed by Western blot analysis using DYKDDDDK antibodies (Rabbit). The relative enrichment of RNAs was assessed by RT-qPCR, and normalized to the levels of spike in Fluc RNA in each sample. p -value = 0.02 with Student's t test comparing to the depleted motif sample.

3) Relatedly, as potential functional validation of the identified proteins, the authors used a modified tethered-function assay where an engineered PUF protein domain was used to target proteins of interest to designed circRNAs that encode GFP. Here they observed the specific increase in GFP translation when PABPC1, and more modestly HNRNPU, was tethered. This finding recapitulated the effect of the “neo-activator” on cyclin-B1 mRNA reported in Figure 6 of Campbell et al (<https://www.nature.com/articles/nsmb.2847>), but now in the context of a circRNA, which potentially supports their model. However, the more compelling experimental validation where the authors tested whether overexpression of PABPC1 could enhance circRNA translation – without tethering – is missing important controls and/or experimental details. Unless I’ve missed it, the authors only examined the effect of PABPC1 overexpression on a circRNA with the A10 sequence. To ensure specificity of the enhancement effect, the authors should examine whether PABPC1 overexpression enhances translation of a circRNA with a depleted sequence. Similarly, it is unclear how loading of the western blots was normalized, making it difficult to assess whether PABPC1 overexpression also increased translation of the GAPDH loading control.

We indeed applied the PUF-tethering approaches routinely used in our lab and Marvin Wickens’ lab to study RBP function, and have now included this paper in our citation (ref 40 in the main text). The reviewer also made a good suggestion to add a specificity control in the experiment of PABPC1 overexpression to enhance circRNA translation without tethering. As suggested, we repeated this experiment with a negative control element (G)₁₀ that do not promote circRNA translation, and found that the activity of PABPC1 and PABPC4 are indeed specific to the IRES-like short sequence (A)₁₀.

In terms of the normalization during quantification of protein levels, all GFP protein was first normalized to the GAPDH level, and then the relative amount of GFP translation was calculated by dividing the mock transfection. This information and the new data were included as the updated Fig. 3E in revised manuscript (page 13).

Fig. 3E. Validation of PABPC1 activity. The expression vector of PABPC1 and various control RBPs were co-transfected with circRNA translation reporter containing (A)₁₀ or (G)₁₀ sequences before the start codon. The protein products were assayed at 48 hours after transfection. The bar graph represents the quantification of GFP levels relative to GAPDH. The protein levels were also normalized to the RNA (n = 3, mean ± SD).

Minor points:

1. “Interestingly, the deletion of “AUC” sequence reduced translation in 4 out of 6 enriched hexamers tested but increased translation driven by one hexamer (poly-A

sequences).” Is this an effect of Kozak context on translation efficiency? The “AUC” would provide optimal -3 context (the A). However, related to the point above, that same A would complete the canonical AAUAAA cleavage & polyadenylation signal, particularly for the “UAAUAA” hexamer sequence, where the largest decrease in translation is observed upon deletion of the spacing “AUC”. To discriminate these potential effects, the authors could substitute the A in the spacing “AUC” for G, which would provide near optimal Kozak context.

We thank the reviewer for the constructive suggestion. This experiment was suggested by a previous reviewer during another submission, who was wondering if the adjacent sequence may affect our screen results. Our original experiments with ATC deletion suggested the activity of enriched elements were not dependent on the “ATC” adjacent sequences.

However, the “ATC” context somewhat resembles the Kozak sequence, and also provided optimal -3 context (the A), which may complicate the data interpretation. As suggested by this reviewer, we have substituted the “AUC” with the “GUC” (a near optimal Kozak context), and found that most enriched hexamers still drive circRNA translation (see below in rebuttal Fig. R4). However, the effects of the new context on circRNA could not simply be attributed to the translation enhancement by Kozak context (i.e., the inclusion of GUC had increased translation in 2 hexamers and decreased translation in another 4 hexamers). The main conclusion for this experiment is that the screen results were not dominated by the adjacent context, however the adjacent context may have some profound effect on the activity of different IRES-like elements. To avoid confusion from our main point, we deleted the last sentence of this paragraph from page 7, because this observation only reflected a specific sequence context that is unrelated to Kozak sequence.

Fig. R4. Effects of neighboring sequence on circRNA translation. The enriched and depleted hexamers were inserted into circRNA reporters with or without ATC trimer (partially resemble Kozak sequence), and were transfected into 293T cells. The samples were analyzed using the same condition as panel E. The circRNA reporters inserted with poly-A sequence were loaded twice as control in both blots.

In addition, we want to mention that a random screen like this has to be performed using a reporter with certain sequence context, and therefore we extracted the enriched consensus motifs from the obtained sequences and then directly tested the

activity of these enriched motifs. We think the original AUC context do not cause the cleavage & re-polyadenylation of circRNA, mainly because of the design of our screen (please see our response of the first point).

Reviewer #3 (Remarks to the Author):

Fan et al., presents an interesting work on circRNAs. The authors argue that circRNAs-coded peptides are actually translated, and they are cell type, tissue, or biological process specific. To prove their presence, the authors adopted mass spec-trometry based peptide identification, on which my review will focus. The datasets used for the search are two published human datasets. And the database was constructed from the three frames of the published circRNAs sequences. Peptides were searched using pFind with FDR threshold (or q-value threshold) of 1%. Further stringent filtration to prevent false peptide matches from non-circRNA region was also used. After the search, more than 2,000 peptides were identified and analyzed.

Overall, the peptide ID procedure is straight and considerably stringent. I do not see much possibility that those 2,000 peptides represent false positives or those from non-circRNA area. However, the authors could address the following points for better understanding of MS-based readers.

We thank the reviewer for appreciating the novelty and broad interests of this work, and have addressed the specific concerns with additional analyses.

1) It would be nice if the authors specify a) the size of datasets (#spectra), b) the size of constructed circRNA DB along with the appended human DB (in terms of #proteins or length of amino acid sequences). They could be on Methods section.

We thank the reviewer for this suggestion, and have now included the information in page 9 on Supplementary Methods. Briefly, *"We used pFind (with default search that not allowed blind modification) to search two previous published human comprehensive proteome datasets (22,909,431 spectra) against a combined database containing all UniProt human proteins (174,392) and the potential circRNA-coded peptides (177,907) across back splice junctions from all three frames."*

2) In supplement table 4, it would be nice to add the explanation on the remaining columns. Some are explained but others are not. What is the last column meaning (separate FDR)?

We thank the reviewer for pointing out the missing information, and have included the explanation of separate FDR in last column of Table S4. We also included detailed explanation on FDR calculation in the Table S4 legend (the first tab of the excel file). *"Separate FDR: the PSMs (peptide-spectrum matches) of decoy peptide are used separately to estimate the false positive discover for the targeted circRNA-coded candidates. For each PSMs of circRNA-coded peptides, the corresponding separate FDR is defined by the total number of above-threshold decoy PSMs (the final score of the peptide) divided by the number of above-threshold target PSMs."*

3) Maybe I missed, but it is not well described that the ID from human protein DB are filtered out (only circRNA hits were retained).

Indeed we selected the spectra that only have circRNA hits, and the reviewer is correct in his/her interpretation. We have now added more detailed information on the Methods section (page 9 of supplementary method), and also included such information below: *“We selected positive mass spectra across back splice junction using following thresholds: $q < 0.01$ (spectrum level), peptides length ≥ 8 with a new sequence of at least 2 aa at either side of the back splice junction, missed cleavage sites ≤ 3 , allowing only common modifications (cysteine carbamidomethylation, oxidation of methionine, protein N-terminal acetylation, pyro-glutamate formation from glutamine, and phosphorylation of serine, threonine, and tyrosine residues).”*

4) Out of 2,000 ID spectra, how many were identified in the previous publication? If this number is low, proteomics readers would be very intrigued because such non-standard peptides are always of a great interest for them (often called dark matter..).

The reviewer is correct that the number of peptides encoded by circRNA in the previous publication is low⁴, and > 90% of our newly identified spectra are not reported previously. We have now included this information in the revised text (page 15-16). We totally agree with the reviewer that it is very intriguing to detect ‘dark matter’ in MS data^{5,6}, which is also one of our research interests. Meanwhile, the levels of these non-standard peptides are generally lower than these from standard proteins, which make reliable identification of these peptides a technically difficult but biologically important question. We have included new text in the discussion to echo the point of this reviewer (page 30-31).

5) pFind enables blind modification search as far as I know. Is this search also allowed blind modification?

We used the “default” search of pFind that allows a maximum of one unexpected modification from a modification list for each peptide. This is more stringent than the “blind” search that do not relay on the known modification list. Not including blind modification may increase the specificity of circRNA-coded peptide discovery but will reduce the sensitivity. In another word, more circRNA encoded peptides may be discovered if we allow the blind modification, at the cost of higher false discovery rate.

We have now included the information in page 9 on Supplementary Methods. Briefly, *“We used pFind (with default search that not allowed blind modification) to search two previous published human comprehensive proteome datasets...”*

6) The annotations shown in Fig 4 a) look almost perfect, and delivered the point of the paper very well. But it would be nice if some more examples could be shown in the supplement figures.

I guess what the reviewer meant is the Fig. 5A, where selected examples of MS spectra for circRNA-coded peptide were shown. We chose some “good” examples for following-up testes, and as the reviewer suggested, we have now included four

additional examples in Fig. S6 that do not look as perfect as the other examples.

7) is q-value used in search: is it peptide level or spectrum level?

The q-value used in search is spectrum level, we have now included the information in the supplementary methods (page 9): “We selected positive mass spectra across back splice junction using following thresholds: \$q < 0.01\$ (spectrum level)...”

Overall, this paper looks very interesting and the analysis was done very well, in particular in terms of MS-based peptide analysis.

We thank the reviewer for the appreciation of this work.

Reference:

1. Wang, Y. & Wang, Z. Efficient backsplicing produces translatable circular mRNAs. *RNA* **21**, 172-9 (2015).
2. Chen, C.K. et al. Structured elements drive extensive circular RNA translation. *Mol Cell* **81**, 4300-4318 e13 (2021).
3. Li, X. et al. Coordinated circRNA Biogenesis and Function with NF90/NF110 in Viral Infection. *Mol Cell* **67**, 214-227 e7 (2017).
4. Kristensen, L.S. et al. The biogenesis, biology and characterization of circular RNAs. *Nat Rev Genet* **20**, 675-691 (2019).
5. Skinner, O.S. & Kelleher, N.L. Illuminating the dark matter of shotgun proteomics. *Nat Biotechnol* **33**, 717-8 (2015).
6. Kong, A.T., Leprevost, F.V., Avtonomov, D.M., Mellacheruvu, D. & Nesvizhskii, A.I. MSFragger: ultrafast and comprehensive peptide identification in mass spectrometry-based proteomics. *Nat Methods* **14**, 513-520 (2017).

REVIEWERS' COMMENTS

Reviewer #1 (Remarks to the Author):

All my previous comments have been addressed. I do not have further comments.

Reviewer #2 (Remarks to the Author):

The authors have addressed my concerns in the revised manuscript.

Reviewer #3 (Remarks to the Author):

The authors addressed most of the points raised in the last round of revision regarding the peptide identification via mass spectrometry. Not only the details on search methods and parameters, but also additional annotation examples are expected to raise the quality of this manuscript and make their arguments more convincing.

The one point that I want re-ask a question about is the point 4) which is about previous identification results. My question was about how many spectra that identified in this study as circRNA hits were also identified in the previous study. This is not about "which circRNAs were identified in the previous study" but about "out of the spectra that are identified in this study as circRNAs, how many were identified in the previous study."

One MS2 spectrum is supposed to be from a single analyte whether it is from a peptide or a circRNA, at least with a high chance. For sure, so called "chimera" MS2 spectrum containing multiple species of different analytes exists but they are quite rare as compared with normal spectra. Thus, when the authors identified 2,000 spectra as circRNA, those specific spectra should not be identified as peptides or other analytes in theory. So I wonder if those 2,000 spectra have been identified in the previous study, the one the authors took the dataset from. Or the authors could search this spectra against human proteome database only to see these 2,000 are indeed not matched to human peptides well.

Reviewer #3 (Remarks to the Author):

The authors addressed most of the points raised in the last round of revision regarding the peptide identification via mass spectrometry. Not only the details on search methods and parameters, but also additional annotation examples are expected to raise the quality of this manuscript and make their arguments more convincing.

We thank the reviewer for the suggestions and recognizing the efforts in our revision.

The one point that I want re-ask a question about is the point 4) which is about previous identification results. My question was about how many spectra that identified in this study as circRNA hits were also identified in the previous study. This is not about "which circRNAs were identified in the previous study" but about "out of the spectra that are identified in this study as circRNAs, how many were identified in the previous study."

One MS2 spectrum is supposed to be from a single analyte whether it is from a peptide or a circRNA, at least with a high chance. For sure, so called "chimera" MS2 spectrum containing multiple species of different analytes exists but they are quite rare as compared with normal spectra. Thus, when the authors identified 2,000 spectra as circRNA, those specific spectra should not be identified as peptides or other analytes in theory. So I wonder if those 2,000 spectra have been identified in the previous study, the one the authors took the dataset from. Or the authors could search this spectra against human proteome database only to see these 2,000 are indeed not matched to human peptides well.

We are sorry for the confusing reply to the original question. Our analysis pipeline was designed to filter out the known peptides previously identified by others and find the remaining peptides matching to the back-splicing junctions. Therefore the 2000 spectra reported here has essentially no overlap with the known peptides from previous studies, including all potential splicing isoforms and noncanonical proteins. In another word, the reviewer is correct in that these spectra were not identified as peptides or other analytes by other studies.

The filters used in our analysis were originally included in both results and methods. These filters ensure that the ~2000 identified spectra have minimal overlap with peptides identified in previous studies. To avoid confusion, we have now moved all details in the same paragraph of methods (page 36). *"(i) We combined a database containing all UniProt human proteins and potential circRNA-coded peptides from all three frames, and searched two comprehensive human proteome datasets (22,909,431 spectra) against this combined database. The spectra mapped to known human proteins in UniProt were removed. (ii) For the remaining spectra that mapped to circRNA-coded peptides, we used Blastp 2.6.0+ to further remove the peptides that are homologues to known proteins (allowing two mismatches) in non-redundant human protein database. (iii) We finally used a strict cutoff to select positive spectra in which the circRNA-coded peptides were broken into fragment ions at both sides of back splice junction."*

We agree with the reviewer on the implication of such results on the 'dark matter' of MS data. Our next research goal is to develop better methods to reliably identify these peptides, which is technically difficult but biologically important. We want to thank the reviewer to bring up this point.